# Daytime variation of aerosol indirect effect for warm marine boundary layer clouds in the eastern north Atlantic

Shaoyue Qiu[1], Xue Zheng[1], David Painemal[2,3], Christopher R. Terai[1], and Xiaoli Zhou[4,5]

[1]Atmospheric, Earth and Energy Division, Lawrence Livermore National Laboratory, Livermore, California, USA
[2] Science Directorate, NASA Langley Research Center, Hampton, VA, USA
[3] Analytical Mechanics Associates, Hampton, VA, USA
[4]Chemical Sciences Laboratory, NOAA, Boulder, CO, USA,
[5]Cooperative Institute for Research in Environmental Sciences (CIRES), University of Colorado, Boulder, CO, USA

*Correspondence to*: Shaoyue Qiu (qiu4@llnl.gov)

Submit to Atmospheric Chemistry and Physics            July 21, 2023.

**Abstract.** Warm boundary layer clouds in the Eastern North Atlantic region exhibit significant diurnal variations in cloud properties. However, the diurnal cycle of the aerosol indirect effect (AIE) for these clouds remains poorly understood. This study takes advantage of recent advancements in the spatial resolution of geostationary satellites to explore the daytime variation of AIE by estimating the cloud susceptibilities to changes in cloud droplet number concentration ($N_d$). Cloud retrievals for four months of July (2018-2021) from SEVIRI on Meteosat-11 over this region are analyzed. Our results reveal a significant "U-shaped" daytime cycle in susceptibilities of cloud liquid water path (LWP), cloud albedo, and cloud fraction. Clouds are found to be more susceptible to $N_d$ perturbations at noon and less susceptible in the morning and evening. The magnitude and sign of cloud susceptibilities depend heavily on the cloud state defined by cloud LWP and precipitation conditions. Non-precipitating thin clouds account for 44% of all warm boundary layer clouds in July and they contribute the most to the observed daytime variation. Non-precipitating thick clouds are the least frequent cloud state (10%), they exhibit more negative LWP and albedo susceptibilities compared to thin clouds. Precipitating clouds are the dominant cloud state (46%), but their cloud susceptibilities show minimal variation throughout the day.

We find evidence that the daytime variation of LWP and albedo susceptibilities for non-precipitating clouds are influenced by a combination of the diurnal transition between non-precipitating thick and thin clouds and the "lagged" cloud responses to $N_d$ perturbations. The daytime variation in cloud fraction susceptibility for non-precipitating thick clouds can be attributed to the daytime variation in cloud morphology (e.g., overcast or broken). The dissipation and development of clouds do not adequately explain the observed variation in cloud susceptibilities. Additionally, daytime variation of cloud susceptibility is primarily driven by variation in the intensity of cloud response rather than the frequency of occurrence of cloud states. Our results imply that polar-

orbiting satellites with overpass time at 13:30 local time underestimate daytime mean value of cloud susceptibility,

as they observe susceptibility daily minima in the study region.

## 1. Introduction

Warm boundary layer clouds, including stratus, stratocumulus, and cumulus clouds, are prevalent over the sub-tropical oceans, account for over 30% of the global annual mean cloud coverage (Warren et al., 1988; Wood, 2012). These clouds have a significant net negative radiative forcing on the surface radiation budget. However, our understanding of the aerosol indirect effect (AIE) on these clouds, particularly the impact of aerosols on cloud amount, brightness, and lifetime, remains a significant source of uncertainty in estimating the radiative forcing from human activities. The AIE plays a critical role in the Earth's radiation budget through its interactions with clouds. It consists of two effects: the Twomey effect, which involves the increase in cloud droplet number from increasing aerosols, and leads to an increase in cloud albedo ($\alpha_c$) from smaller droplets when the cloud liquid water path (LWP) is held constant (Twomey, 1977), and the cloud adjustment effect, which encompasses the impact of aerosols on cloud amount, cloud water, and $\alpha_c$ through modulating cloud processes (e.g., Albrecht, 1989; Xue and Feingold, 2006; Chen et al., 2014; Gryspeerdt et al., 2019). The cloud adjustment effect is highly variable with large uncertainties in signs and magnitudes depending on cloud state, boundary layer, and meteorological conditions among other factors (e.g., Han et al., 2002; Wang et al., 2003; Small et al., 2009; Sato et al, 2018).

Previous studies have made significant progress in identifying different cloud processes and feedback mechanisms to explain the responses of CF, LWP, and $\alpha_c$ to aerosol perturbations (e.g., as summarized in Steven and Feingold, 2009; Fan et al., 2016; Gryspeerdt et al., 2019). The cloud adjustment effect is influenced by two key feedback mechanisms: precipitation suppression, and sedimentation-evaporation-entrainment.

Under clean conditions and for clouds predominantly precipitating, an increase in the cloud droplet number concentration ($N_d$) and associated decrease in droplet sizes, reduces precipitation efficiency, and decreases water loss from precipitation. Consequently, this promotes an increase in cloudiness and cloud LWP (Albrecht, 1989; Qian et al., 2009; Li et al., 2011; Terai et al., 2012, 2015). For non-precipitating clouds, decreased cloud drop size due to increases in $N_d$ impacts CF and LWP through their impact on the entrainment rate. A decrease in cloud droplet size diminishes the sedimentation rate in clouds, causing an accumulation of cloud water near the cloud top. This increased cloud water in the entrainment zone enhances cloud-top radiative cooling, entrainment rate, and evaporation, resulting in a decrease in CF and cloud LWP (Bretherton et al., 2007; Chen et al., 2014; Toll et al., 2019; Gryspeerdt et al., 2019).

Additionally, the faster evaporation rates from smaller droplets enhance cloud-top cooling, downward motion in clouds, total kinetic energy, and horizontal buoyancy gradient. The processes listed above, in turn, increase evaporation and entrainment rate and, thus, forming a positive feedback loop (Wang et al., 2003; Xue and Feingold, 2006; Small et al., 2009; Toll et al., 2019). Furthermore, among non-precipitating clouds, thick clouds with larger LWP exhibit stronger cloud-top longwave radiative cooling rate and therefore stronger cloud-top entrainment rate (e.g., Sandu et al., 2008, Williams and Igel, 2021). Therefore, the classification of cloud states (e.g., precipitating conditions and thickness) is essential for accurately quantifying the AIE and discerning opposing cloud processes. In this study, we classify cloud states based on the LWP-$N_d$ parameter space, as these variables provide the most informative metrics for cloud susceptibility (Zhang et al., 2022).

This study focuses on the Eastern North Atlantic (ENA) region, where the U.S. Department of Energy
(DOE) Atmospheric Radiation Measurement program (ARM) deployed the ground-based user facility at the Azores
archipelago (Mather and Voyles, 2013). During the summer over ENA region, warm boundary layer clouds exhibit
pronounced diurnal variations in their properties and cloud states. For example, based on ARM surface radar and
lidar observations, the frequency of stratocumulus clouds is highest at night, accompanied by an increase in the
fraction of precipitating clouds. Throughout the daytime, both cloud fraction and precipitation fraction experience a
slight decrease, followed by an increase after sunset (Remillard et al, 2012). The retrieved cloud microphysical
properties from ARM ground-based observations show similar "U-shaped" diurnal variations in cloud LWP, liquid
water content, and optical thickness (Dong et al., 2014). Additionally, numerical studies have revealed a distinct
diurnal cycle of AIE for marine stratocumulus clouds, attributed to changes in cloud properties and boundary layer
thermodynamic conditions (e.g., Sandu et al., 2008, 2009). However, observational analyses based on the ground-
based observations at the ENA site or in-situ measurements from field campaigns are often based on a few cases
with limited samples and insufficient spatial coverage (e.g., Liu et al., 2016; Wang et al., 2021; Zheng et al., 2022).
There have been few observational studies investigating the diurnal cycle of AIE in the ENA region. With recent
advancements in the spatial resolution of geostationary satellites, this study aims to investigate the diurnal variation
of the AIE in warm boundary layer clouds over the ENA region and gain a better understanding of the underlying
mechanisms.
Both cloud properties and meteorological conditions have substantial spatiotemporal variability and distinct
diurnal variations. Furthermore, changes in meteorological conditions can in turn influence cloud and aerosol
properties. One of the main challenges in understanding the AIE lies in isolating the impacts of the confounding
meteorological drivers on clouds and aerosols from AIE on clouds. To address this challenge, Gryspeerdt et al.
(2016) proposed the use of $N_d$ as an intermediary variable for AIE, instead of using aerosol optical depth (AOD) or
aerosol index. The use of $N_d$ circumvents the well-known dependency of AOD on CF and surface wind speed,
which does not necessarily reflect actual changes in aerosol loading. Moreover, the control of relative humidity and
aerosol type on AOD prevents to establish a direct link between AOD and aerosol concentration or cloud
condensation nuclei (CCN).
Another common method to disentangle meteorological impacts is to sort the controlling meteorological
factors of cloud state, such as relative humidity, lower tropospheric stability, vertical velocity, and examine the AIE
accordingly (e.g., Chen et al., 2014; Gryspeerdt et al., 2019). However, this approach overlooks important
information, including the frequency of occurrence of specific environmental conditions, the spatiotemporal co-
variation of meteorological factors, and the correlations among them. Zhou et al. (2021) and Zhang et al. (2022)
proposed a new method to estimate the cloud susceptibility within a confined space (e.g., a 1° × 1° or 2° × 2° grid
box) of each satellite snapshot by assuming consistent meteorological conditions within this spatial domain.
Additionally, it is important to note that meteorological conditions influence albedo susceptibility by altering the
frequency of occurrence of different cloud states (e.g., precipitating and non-precipitating). Specifically, within a
particular cloud state, meteorological conditions offer limited information regarding cloud susceptibility (Zhang et
al, 2022).

The second main source of uncertainty in observational AIE studies arise from inferring processes in a

temporally evolving system based on snapshots of observations (Mülmenstädt and Feingold, 2018). Due to the
limited temporal or spatial resolution of the observations, most studies assume a Markovian system, where clouds
and AIE are assumed to only relate to the current state of the system and have no memory of the past states.
However, this assumption contradicts the nature of the cloud system. Observational and modeling studies have
shown that aerosol-cloud interaction processes take hours to reach the equilibrium state and the sensitivity of AIE is
time dependent. For instance, Glassmeier et al. (2021) applied a Gaussian-process emulation and derived the
adjustment equilibration timescale for LWP to be ~20 hours. By tracking the ship tracks in satellite observations,
Gryspeerd et al. (2021) found a similar AIE timescale of ~20 hours or longer and the magnitude of LWP
susceptibility increases with time. In addition, Christensen et al. (2020) discovered that influence of aerosols on
cloud LWP, CF, and cloud top height persists two to three days by tracking cloud systems in satellite observations.
In summary, the sensitivity of cloud responses to $N_d$ perturbations changes with time and, thus, the assumption that
AIE has no memory of its past state is inadequate. Nonetheless, the direct evaluation of the impact of cloud memory
on the quantified cloud susceptibility remains unexplored, to the best of our knowledge.

To facilitate a process-level understanding of the drivers behind the diurnal variation of AIE for warm

boundary layer clouds, we will classify these clouds into three states: precipitating clouds, non-precipitating thick
clouds, and non-precipitating thin clouds. We investigate the changes in both the frequency of occurrence of cloud
states and the magnitude of AIE for different cloud states throughout the day. Additionally, we document the
temporal changes in cloud state within each fixed $1° \times 1°$ grid box and quantify the influences of cloud memory and
state transition on AIE. Section 2 describes the datasets as well as the methodology employed to quantify cloud
susceptibilities, distinguish precipitating clouds from the satellite retrievals, and track cloud states. We present our
results in Section 3. Section 3.1 characterizes the general conditions of warm boundary clouds over the ENA region
during the summer. Section 3.2 introduces the LWP-$N_d$ parameter space and illustrates the dependence of cloud
responses to $N_d$ perturbations on cloud states. We then discuss the mean daytime variation of cloud susceptibilities
for all cloud states in Section 3.3, followed by an analysis of the AIE daytime variation for each cloud state and the
impact of the state transition on AIE in Section 3.4. In Section 3.5, we decompose the contributions to the daytime
variation of cloud susceptibility into two components, one is from changes in the frequency of occurrence of
different cloud states and the other is from changes in the intensity of AIE during the day. Section 4 includes
discussions on the similarities and differences in findings between this study and previous studies of AIE and
Section 5 is the summary and conclusions of this study.
**2. Dataset and Methodology**

We use cloud retrievals derived from the Spinning Enhanced Visible InfraRed Imager (SEVIRI) on

Meteosat-11, with a spatial resolution of 3 km at nadir and a half-hourly temporal resolution over the ENA region
(33-43°N, 23-33°W). SEVIRI cloud products are derived using the Satellite ClOud and Radiation Property retrieval
System (SatCORPS) algorithms (e.g., Painemal et al., 2021), based on the methods applied by the Clouds and the
Earth's Radiant Energy System (CERES) project, and specifically tailored to support the ARM program over the
ARM ground-based observation sites (Minnis et al. 2011, 2020). Given the purpose of this study on quantifying the
AIE on warm boundary layer clouds, we focus on four months of July (2018-2021), a period that coincides with the
highest frequency of occurrence of warm boundary layer clouds over the ARM ENA site (Rémillard et al. 2012;
Dong et al., 2014, 2023).
The cloud mask algorithm implemented in SatCORPS is described in Trepte et al. (2019). SatCORPS cloud
properties are based on the shortwave-infrared split-window technique during daytime (VISST, Minnis et al. 2011,
2020), with cloud optical depth ($\tau$) and effective radius ($r_e$) being derived using an iterative process that combines
reflectance and brightness temperatures from the 0.64 μm and 3.9 μm channels. Cloud LWP is computed from $\tau$ and
$r_e$ using the formula $LWP = \frac{4 r_e \tau}{3 Q_{ext}}$, where $Q_{ext}$ represents the extinction efficiency and assumed constant of 2.0
(Minnis et al. 2011, 2020). The top-of- atmosphere (TOA) broadband shortwave $\alpha_c$ is derived from an empirical
radiance-to-broadband conversion using the satellite imager's visible channel and CERES Single Scanning Footprint
(SSF) shortwave fluxes, and dependent on solar zenith angle and surface type (Minnis et al. 2016). Cloud top height
computation follows the methodology in Sun-Mack et al. (2014).
To validate the Meteosat-11 retrieved cloud mask and the detection of boundary layer clouds, we compare
the boundary layer cloud fractions derived from Meteosat-11 with the ground-based observations at the ARM ENA
site. As seen in Fig.S1, both the diurnal variation and the mean CF of Meteosat-11 agree well with ARM
observations. More details on the methodology for the evaluation study are included in the supplementary material.
Our analysis focuses on warm boundary layer clouds with cloud tops below 3km and a liquid cloud phase.
To focus specifically on boundary layer cloud cases without including the edges of deep clouds, we apply a stricter
threshold than merely using the pixel-level cloud top height. We define boundary layer clouds as those with 90% of
their cloud tops below 3km, labeling all contiguous cloudy pixels as distinct cloud objects.
Cloud $N_d$ is retrieved based on the adiabatic assumptions for warm boundary layer clouds, as in Grosvenor
et al. (2018) according to the following equation:
$$N_d = \frac{\sqrt{5}}{2\pi k}\left(\frac{f_{ad}c_w\tau}{Q_{ext}\rho_w r_e^5}\right)^{1/2} \tag{1}$$

In Equation (1), $k$ represents the ratio between the volume mean radius and $r_e$, assumed to be constant of 0.8 for
stratocumulus; $f_{ad}$ is the adiabatic fraction of the observed liquid water path and assumed to be 0.8 for
stratocumulus clouds (Brenguier et al., 2011; Zuidema et al., 2012); $c_w$ is the condensation rate, which is a function
of temperature and pressure; $Q_{ext}$ is the extinction coefficient, approximated as 2 in this study; and $\rho_w$ is the density
of liquid water. While the different components of Eq. (1) could contribute to the uncertainties in $N_d$, errors in $r_e$ are
the dominant drivers in Eq. (1) (Grosvenor et al., 2018).
To minimize uncertainties associated with bias in satellite cloud microphysical retrievals, we only select
pixels with a minimum $r_e$ of $3\mu m$, a minimum $\tau$ of 3, and a solar zenith angle (SZA) of less than 65° (e.g., Painemal
et al., 2013; Painemal, 2018; Zhang et al., 2022). The SZA threshold of 65° was chosen to minimize biases observed
at high solar zenith angle in $r_e$ and $\tau$ (e.g., Grosvenor & Wood, 2014; Grosvenor et al., 2018).

In addition, to reduce uncertainties associated with the adiabatic assumption in the $N_d$ retrieval, we

implement a filtering process. For each cloud, we exclude cloudy pixels at the cloud edge, defined as those adjacent
to cloud-free pixels, following a similar sampling strategy suggested by Gryspeerdt et al. (2022). Therefore, all
cloud properties in this study refer to the properties of cloud body without cloud edge. It is worthy of note that
shallow cumulus clouds with diameters smaller than 9km are not included. The removal of cloud-edge pixels
accounts for ~14% of the cloudy pixels. Furthermore, we removed grid boxes containing islands due to the
uncertainties in Meteosat retrievals over contrasting underlying surface. Lastly, to avoid unrealistically large
retrievals, we eliminate pixels with the retrieved $N_d$ values exceeding $1000\ cm^{-3}$, which constituted only 0.002% of
the data.

Cloud susceptibility is quantified as the slope between cloud properties and $N_d$ using a least-square

regression. As found by Arola et al. (2022) and Zhou and Feingold (2023), the retrieved cloud susceptibilities are
sensitive to small-scale cloud heterogeneity, the co-variability between cloud properties and $N_d$, and the spatial scale
of cloud organization. To reduce biases resulting from heterogeneity and co-variability, we first average the 3-km
pixel-level cloud retrievals and $N_d$ (Eq. 1) to a regular $0.25° \times 0.25°$ grid for each half-hourly time step.

To further mitigate the impact from co-variability between cloud properties and $N_d$ at larger spatiotemporal

scales, cloud susceptibility is estimated within a $1° \times 1°$ grid box at each satellite time step (e.g., Zhang et al, 2022).
Moreover, estimating the cloud susceptibility over a confined space also help to constrain the meteorological
impacts on AIE, with the assumption of a homogeneous meteorological condition within this spatial scale. Next,
susceptibilities are calculated using the 0.25° smoothed data if the number of data points within the $1° \times 1°$ box
exceeds six (from a maximum of 16 data points). It is important to note that when computing the $0.25° \times 0.25°$
averaged cloud properties, only data from cloudy pixels are used to ensure that the estimated susceptibility is not
weighted by CF or impacted by satellite artifacts. Lastly, due to the minimal spatial variability of cloud
susceptibility in the study region, the 1° cloud susceptibility is averaged over the study region (33-43°N, 23-33°W)
to characterize the daytime variation of AIE. Additionally, results and conclusions of this study are not sensitive to
the size of the box calculating the cloud susceptibility (e.g., over a $0.8° \times 0.8°$ box or over a $1.5° \times 1.5°$ box, not
shown).

Because of the nonlinear relationships between LWP and $N_d$, the LWP susceptibility is defined as the slope

in logarithmic scale, that is: $dln(LWP)/dln(N_d)$ (e.g., Gryspeerdt et al. 2019). The albedo susceptibility is
estimated as the slope between $\alpha_c$ and $\ln(N_d)$, equivalent to $d\alpha_c/dln(N_d)$ (e.g., Painemal 2018). Lastly, the CF
susceptibility is estimated as $dCF/dln(N_d)$. The mean CF is defined as the fraction of cloudy pixels excluding
cloud edge to the sum of cloudy and clear pixels within each $0.25° \times 0.25°$ box. Due to the highly variable nature of
CF, the variability in the 0.25° CF could arise from quantifying edges or centers of the same cloud layer rather than
$N_d$ perturbations. To assess the potential influence of cloud morphology on the retrieved CF susceptibility, we
excluded any $1° \times 1°$ scene meeting the following three criteria: 1) the difference between the maximum and
minimum 0.25° CF greater than 0.9, 2) the variation in the 0.25° $N_d$ less than $60\ cm^{-3}$, and 3) the 0.25° CF in the
$1° \times 1°$ box sample the same cloud. The 0.9 and $60\ cm^{-3}$ thresholds represent ~45% of the data. With the three
thresholds combined, a total of 17,000 scenes were removed, which accounts for ~24% of the total samples.
Removing these scenes does not change the conclusions of CF susceptibility in this study (not shown), which
demonstrates that cloud morphology, has minimal impact on the retrieved CF susceptibility. Furthermore, as we
removed $N_d$ retrievals at cloud edge where $N_d$ likely suffers large uncertainties, cloudy pixels at cloud edge are set
as clear for consistency in the calculation of the CF susceptibility. Removing the cloud edge decreases the four-
month mean CF for warm boundary layer clouds from 21.6% to 19.0%.
The susceptibility of the shortwave radiative fluxes to $N_d$ ($F_0$) is estimated as the sensitivity of the TOA
shortwave upward radiative flux ($SW_{TOA}^{up}$) to $N_d$ perturbations (e.g., Chen et al. 2014; Painemal 2018; Zhang et al.
2022). The mean $SW_{TOA}^{up}$ over a $1° \times 1°$ grid box is estimated using Eq. (2), with the assumption that the clear-sky
albedo over the ocean is small compared to the cloud albedo:
$$\overline{SW_{TOA}^{up}} = \overline{SW_{TOA}^{dn}} \cdot \overline{\alpha_c} \cdot \overline{CF}, \tag{2}$$

where $SW_{TOA}^{dn}$ is the grid-box mean TOA shortwave downward radiative flux, which is estimated based on the
latitude, longitude, date, and overpass time of each pixel, $\alpha_c$ and CF are the grid-box mean values. Then, $F_0$ is
estimated using the calculated $\alpha_c$ and CF susceptibilities, and the $1° \times 1°$ grid-box mean cloud properties as shown
in the equation below:
$$F_0 = -\frac{dSW_{TOA}^{up}}{dln(N_d)} = -\overline{SW_{TOA}^{dn}} \cdot \left(\frac{d\alpha_c}{dln(N_d)} \cdot \overline{CF} + \frac{dCF}{dln(N_d)} \cdot \overline{\alpha_c}\right). \tag{3}$$

$F_0$ is in the unit of $W\ m^{-2}\ ln\ (N_d)^{-1}$, and a positive value indicates a decrease in the $SW_{TOA}^{up}$, which is a *warming*
effect at the surface.
To minimize uncertainties in the linear regression for the estimated susceptibility, we analyze regressions
that exhibited a goodness of fit exceeding the 95% confidence interval (i.e., $\chi^2 < \chi_{0.95,c}^2$), and an absolute
correlation coefficient greater than 0.2 (e.g., Painemal, 2018; Zhang et al., 2022). There is a total of ~115,000
samples of the 1° cloud susceptibilities in this study, applying the goodness of fit thresholds result in exclusions of ~
33,000- 43,000 samples for different susceptibilities, which are ~28-37% of the data. Sensitivity test shows that
including cases that fail the goodness of fit test will not change the results and conclusions of this study (not shown).
Specifically, including these cases decrease the magnitude of cloud susceptibilities for all three cloud states, but the
signs of cloud responses to $N_d$ perturbations remain consistent.
Since precipitating and non-precipitating clouds exhibit distinct responses to aerosol perturbations due to
the effect of precipitation suppression and the wet-scavenging feedback, it is critical to distinguish between these
two cloud states when estimating AIE. Previous studies have utilized various methods based on the effective radius
threshold (e.g., Gryspeerdt et al., 2019, Toll et al., 2019; Zhang et al., 2022) and the rain rate threshold (e.g., Duong
et al., 2011; Terai et al., 2015) from satellite retrievals. In our study, we validate these two methods using the
precipitating mask estimated from ground-based observations with a radar reflectivity threshold together with the
lidar-defined cloud base at the ARM ENA site (e.g., Wu et al., 2020). The thresholds of $r_e > 12\ \mu m$ and $r_e > 15\ \mu m$
yield hit rates of 0.79 and 0.73, respectively. However, the false alarm rate is higher for $r_e > 12\ \mu m$ (0.21) compared
to $r_e > 15\ \mu m$ (0.1). Rain rate is computed using the empirical relationships derived from ground-based
measurements in Comstock et al. (2004) as $R = 0.0156\ (LWP/N_d)^{1.75}$. Using a threshold of R>0.05 mm/h results
in a hit rate of 0.65. Consequently, we use the $r_e > 15\ \mu m$ threshold to define precipitating clouds in this study.

To investigate the dependences of cloud susceptibility on previous cloud states and quantify the influence

of cloud memory on the estimated cloud susceptibility, we track the historical cloud state over a $1° \times 1°$ grid box for
a two-hour period. During the summer in the study region, low wind conditions prevail in the boundary layer, with
the mean wind speed being less than 10 m/s for 85% of the time and less than 7 m/s for 60% of the time. Therefore,
in most cases, less than half of the clouds exit the grid box within the two hours, allowing us to track the previous
cloud state within the same grid box (i.e., from the Eulerian perspective). The influence of cloud memory is assessed
by comparing the cloud susceptibilities of clouds that undergo a transition in cloud state with those that do not
experience such a transition. Section 3.4 provides more details and discussion on the sensitivity of tracking time and
the influence of advection on our classification.
**3. Results**
**3.1 General cloud conditions and mean cloud responses to $N_d$ perturbations**

In the ENA region, characterized by dominant Bermuda High with its prevailing ridge and zonal synoptic

pattern (Mechem et al., 2018), the summer season gives rise to the annual peak in boundary layer cloud coverage
The monthly mean low-level CF retrieved from Meteosat-11 reaches its maximum of 35% in July, compared to an
annual mean of 17% during the four-year study period. This region represents a typical clean marine condition,
situated far from continental influences, which results in a consistently low $N_d$ compared to polluted marine regions,
such as the northeastern (NE) Pacific near California or the northwestern Atlantic near the Gulf of Maine. In July,
the mean $N_d$ over the ENA region is 65 $cm^{-3}$ with the lower 5th and upper 95th percentile of 15 and 160 $cm^{-3}$,
respectively. The retrieved $N_d$ values in this study closely align with in-situ measurements from the Aerosol and
Cloud Experiments in Eastern North Atlantic (ACE-ENA) field campaign. For instance, the in-situ measured $N_d$ in
July 2017 varied from 25 to 150 $cm^{-3}$, with a mean value of 65 $cm^{-3}$ (e.g., Yeom et al., 2021; Zhang et al., 2021).
Moreover, our satellite retrieved $N_d$ exhibits good agreement with retrievals based on ground-based observations at
the ARM ENA site (e.g., Dong et al., 2014; Wu et al., 2020) and the MOderate resolution Imaging
Spectroradiometer (MODIS, e.g., Bennartz 2007; Bennartz and Rausch 2017).

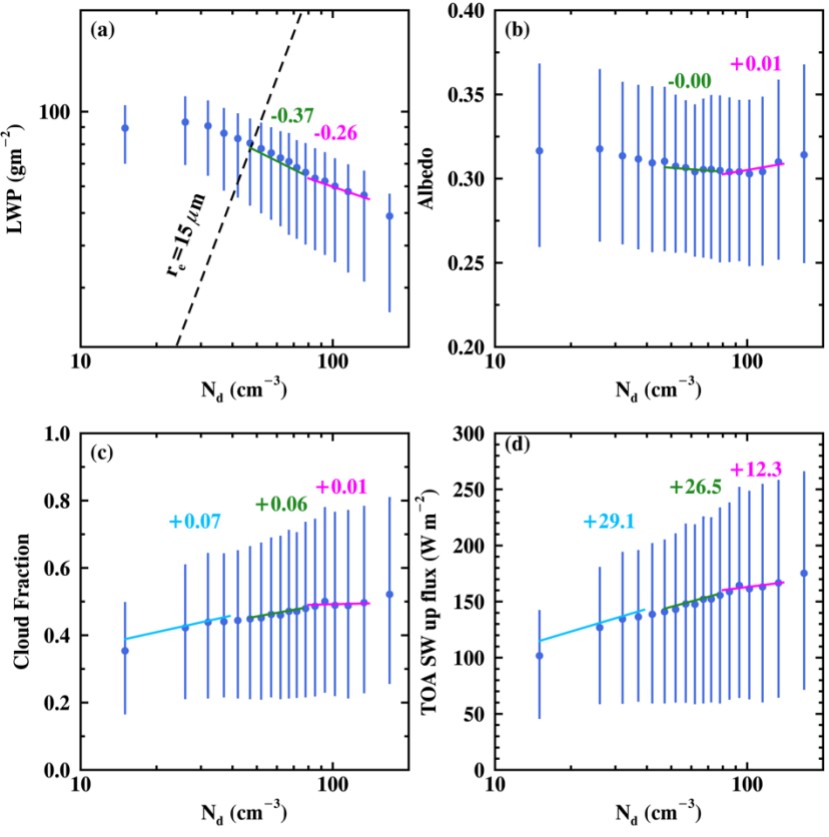

Figure 1. Relationships between $N_d$ and cloud properties: (a) cloud LWP, (b) cloud albedo, (c) cloud fraction, and
(d) TOA shortwave upward radiative flux. The dots represent the mean values, while the whiskers indicate the upper
and lower 25th percentile. In (a), the dashed line denotes $r_e$ =15 $\mu m$, serving as an indicator of precipitation
occurrence, with precipitating clouds located to the left of the line. Blue, green, and magenta lines in panels (a)-(d)
represent the regression slopes of the mean cloud properties, and the mean $ln(N_d)$, for $N_d < 40\ cm^{-3}$, $N_d$ between
40 and 80 $cm^{-3}$, and $N_d > 80\ cm^{-3}$, respectively.
Previous studies have demonstrated that clouds exhibit diverse responses to aerosol perturbations under
clean and polluted conditions (e.g., Fan et al. 2016; Mülmenstädt and Feingold, 2018). Figure 1 shows the
relationships between the climate mean cloud properties, derived from the pixel-level SEVIRI cloud products, and
averaged to the 1° × 1° resolution, as a function of the 1° × 1° mean $N_d$ values. To quantify these responses, cloud
susceptibility is estimated as the slope of the mean cloud variable changes across $N_d$ bins. In pristine conditions ($N_d$
< 40 $cm^{-3}$, ~28% of data), clouds predominantly precipitate ($r_e$ >15 $\mu m$, Fig.1a). The mean cloud LWP features a
slight increase followed by a decrease with increasing $N_d$. This result departs from the precipitation suppression
hypothesis, in which LWP typically increases. The absence of a precipitation suppression signal is likely attributed
to the relatively modest precipitation that occurs in this region during summer (e.g., Wu et al., 2020; Zheng and
Miller, 2022), resulting in a minimal precipitation suppression effect and a dominant entrainment drying effect. In
terms of $\alpha_c$, the potential decrease in $\alpha_c$ attributed to a LWP reduction offsets the potential increases in $\alpha_c$ caused by
the Twomey effect, resulting in a net zero change in mean $\alpha_c$ for clouds with $N_d < 40\ cm^{-3}$ (Fig.1b). Furthermore,
the majority of precipitating clouds are broken, with a mean CF that increases with $N_d$ from 0.35 to 0.45 (Fig.1c).
Consequently, the mean $SW_{TOA}^{up}$ flux increases from 100 to 140 $W\ m^{-2}$ as $N_d$ increases from 10 to 40 $cm^{-3}$. This
increase in CF for precipitating clouds aligns with previous study over the north Atlantic region across all seasons
(e.g., Gryspeerdt et al., 2016). In summary, despite the slight decrease in mean LWP with increasing $N_d$ for
precipitating clouds, the mean cloud albedo remains relatively constant, while the mean CF increases, resulting in an
overall increase in the TOA reflected shortwave flux.

Under relatively polluted conditions with $N_d > 40\ cm^{-3}$ (~72% of data), the mean LWP shows a

decreasing trend with $N_d$. For $N_d$ values between 40-80 $cm^{-3}$, the $ln(\text{LWP})$- $ln(N_d)$ slope is $-0.37$, while for $N_d$
exceeding 80 $cm^{-3}$, the slope reaches $-0.26$ (green and magenta lines in Fig.1a). This negative adjustment of LWP
for non-precipitating clouds is consistent with the sedimentation-evaporation-entrainment feedback, as well as with
previous studies of stratocumulus clouds in other regions (e.g., Gryspeerdt et al., 2019; Zhang et al., 2022). The
mean $\alpha_c$ remains nearly constant within the $N_d$ range of 40-80 $cm^{-3}$ (Fig.1b). As LWP decreases at a slower rate
for $N_d > 80\ cm^{-3}$, the Twomey effect becomes more dominant and leads to a slight increase in $\alpha_c$ with a slope of
0.01 (magenta line in Fig.1b). For non-precipitating clouds, the mean CF slightly increases with increasing $N_d$ with
a CF susceptibility of 0.06 and 0.01 (green and magenta lines in Fig.1c). As a result, the $SW_{TOA}^{up}$ flux exhibits a
weaker susceptibility compared to precipitating clouds (Fig.1d).

**3.2 Daytime mean cloud susceptibilities in the LWP-$N_d$ space**

One limitation of the relationships derived from the mean cloud properties with sorted $N_d$ is the

confounding effect from meteorological impacts on cloud properties and cloud susceptibilities. As a comparison,
Fig.2 shows the mean cloud susceptibility estimated within each half-hourly snapshot's $1° \times 1°$ grid box and
averaged in the LWP-$N_d$ parameter space. There are around 72,000-82,000 samples of the 1° cloud susceptibilities
in this study. The number of samples for different cloud susceptibilities are slightly different due to the goodness of
fit test for each regression. We calculate the mean susceptibilities for LWP-$N_d$ bins with more than 100 cloud
susceptibility samples. Blank bins in Fig.2 are bins with less than 100 samples. Figure 2e shows the occurrence
frequency of samples for the LWP susceptibility in Fig.2a.

With the assumption that the meteorological condition is homogeneous in each grid box, the estimated

cloud susceptibilities exhibit much stronger relationships for all cloud variables compared to the climatological
mean adjustment rates shown in Fig.1. The disparities between the two methods suggest that meteorological
confounders tend to obscure the signal of the AIE over the ENA region. Moreover, the cloud responses for both
precipitating and non-precipitating clouds exhibit consistent signs between the half-hourly (Fig.2) and
climatological-mean approaches (Fig.1). This consistency is likely attributed to the confined domain (a $10° \times 10°$)
and the focus on July in this study, which limit the spatial and temporal covariability between cloud properties and
$N_d$. This consistency also demonstrates that the overall cloud responses to $N_d$ perturbations primarily depend on
cloud states (e.g., precipitating conditions and cloud thickness).

The dependence of cloud response on cloud state is illustrated in Fig.2. We define three cloud states*: (1)

*the precipitating clouds ($r_e > 15\ \mu m$),(2) the non-precipitating thick clouds ($r_e < 15\ \mu m$, LWP $> 75\ gm^{-2}$),* and (3)
*the non-precipitating thin clouds ($r_e < 15\ \mu m$, LWP $< 75\ gm^{-2}$),* similar to the definition in Zhang et al. (2022).

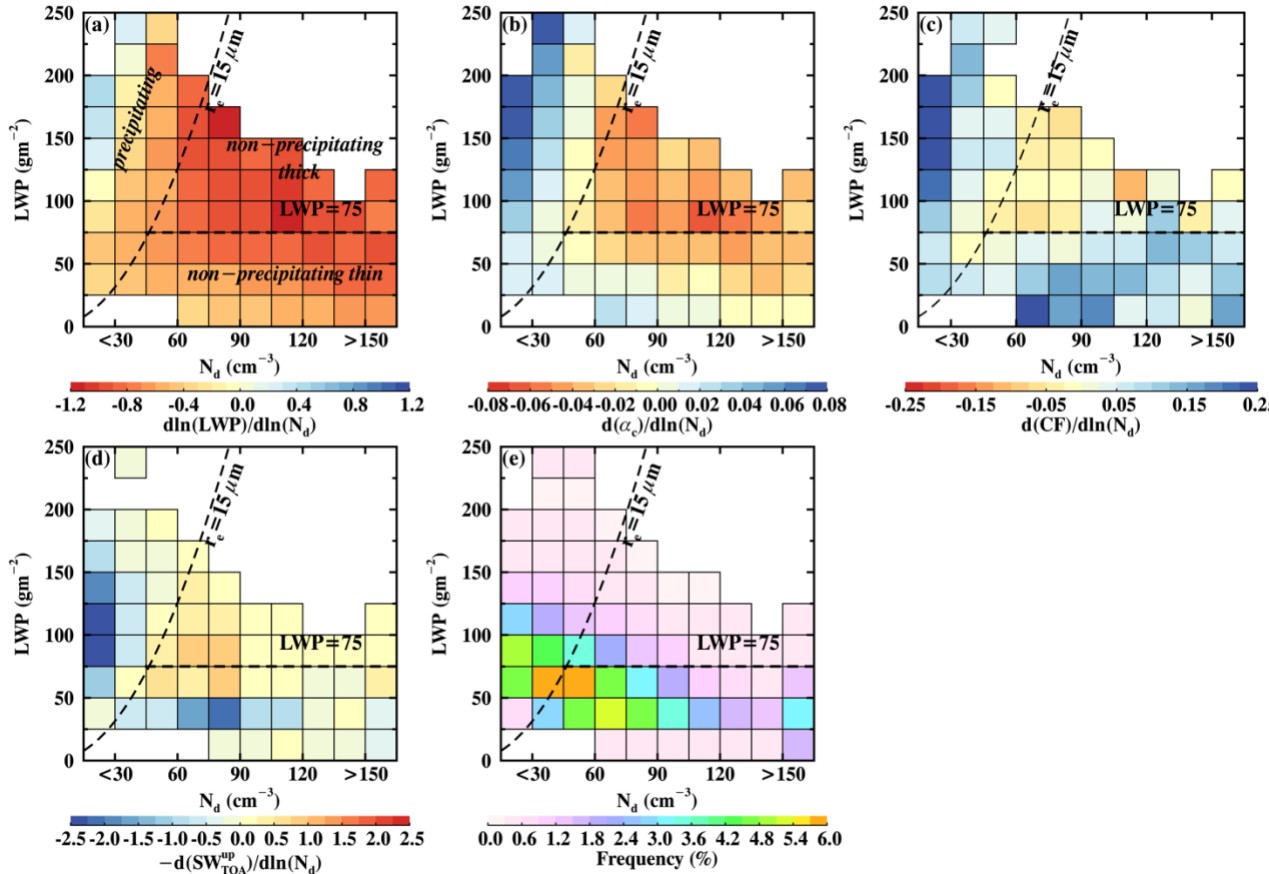

Figure 2. Mean cloud susceptibilities for different $N_d$ and LWP bins during the daytime. (a) cloud LWP susceptibility ($dln(LWP)/dln(N_d)$), (b) cloud albedo susceptibility ($d\alpha_c/dln(N_d)$), (c) cloud fraction susceptibility ($dCF/dln(N_d)$), (d) cloud shortwave susceptibility ($-dSW_{TOA}^{up}/dln(N_d)$) weighted by the frequency of occurrence of samples of each bin, and (e) frequency of occurrence of samples in each bin. The dashed lines in (a)-(e) indicate $r_e = 15\ \mu m$ and LWP= 75 $gm^{-2}$, as thresholds for precipitation (precipitating clouds located to the left of the line) and thick clouds (with LWP > 75 $gm^{-2}$). The defined three clouds states are noted in (a).

*a. Precipitating clouds*

Among warm boundary layer clouds, precipitating clouds are the dominant cloud state in July over the study region, with a total frequency of occurrence of 46% (Fig.2e). The increase in cloud LWP with increasing $N_d$ is observed primarily in heavily precipitating thick clouds with $N_d < 30\ cm^{-3}$ and LWP > 125 $gm^{-2}$ (Fig.2a). However, these clouds occur relatively infrequently at ENA, accounting for only 2% of the total warm boundary cloud population (Fig.2e). In contrast, most of the precipitating clouds at ENA are lightly precipitating with $15 < r_e < 20\ \mu m$ (Fig.2e and Fig.S2c) and they exhibit a slight decrease of LWP with $N_d$ (Fig.2a). The mean LWP susceptibility for lightly precipitating clouds ranges from $-0.5$ to $-0.2$ for different bins, with a mean value of $-0.4$. The standard deviations of LWP susceptibility in different LWP-$N_d$ bins vary between 0.4 to 1.2, while the LWP susceptibilities for precipitating clouds are significantly different than other two cloud states at a 95% confidence level. The slight decrease in LWP for lightly precipitating clouds aligns with previous findings over the Pacific, Atlantic, and global oceans for marine stratocumulus (e.g., Fig S4 in Zhang and Feingold, 2023).

The contrasting response of LWP to $N_d$ perturbations for lightly and heavily precipitating clouds can be
attributed to the interplay of two competing processes: the depletion of LWP caused by the sedimentation-
evaporation-entrainment feedback and the accumulation of LWP resulting from the precipitation suppression
feedback. Heavily precipitating clouds are predominantly overcast with a mean CF of 0.65 (Fig.S2a) and a mean $r_e$
of 25 $\mu m$ (Fig.S2c). Precipitation acts to stabilize the boundary layer, remove water from cloud top, and reduce the
entrainment rate (Sandu et al., 2007, 2008). Therefore, heavily precipitating clouds exhibit smaller entrainment rate
than non-precipitating clouds with similar LWP. The increase of LWP from precipitating suppression feedback
outweighs the decrease of LWP from entrainment feedback and results in a net increase in LWP (e.g., Chen et al.,
2014; Toll et al., 2019). In lightly precipitating clouds, however, the suppression effect of drizzle on the entrainment
rate is minimal. Therefore, the decrease in LWP from entrainment overpowers the increases in LWP from
precipitating suppression, leading to a net decrease in LWP with increasing $N_d$ (e.g., Xue and Feingold, 2006).
Precipitating clouds generally exhibit brighter cloud albedo with increasing $N_d$ as a result of the weak
negative and positive LWP adjustment, particularly in heavily precipitating clouds (Fig.2b). For lightly precipitating
clouds, $\alpha_c$ susceptibilities range from $-0.04$ to 0.07 $ln(N_d)^{-1}$, with a mean of 0.02 $ln(N_d)^{-1}$. The suppression of
precipitation by $N_d$ also lead to a significant increase in CF for heavily precipitating clouds, with slopes greater than
0.25 $ln(N_d)^{-1}$. For most of the lightly precipitating clouds, the mean CF exhibits small variation with $N_d$
perturbations, with CF susceptibilities ranging between $\pm 0.025$ $ln(N_d)^{-1}$ (Fig.2c). The standard deviation of the
1° $\alpha_c$ and CF susceptibilities for different precipitating bins ranges between 0.05-0.15 and 0.3-0.6, respectively. The
$\alpha_c$ and CF susceptibilities for precipitating clouds are significantly different than other two cloud states at a 95%
confidence level. Considering the combined effects of increased $\alpha_c$ and CF, the mean radiative response for
precipitating clouds amounts to $-13$ $W\ m^{-2} ln(N_d)^{-1}$, which is a cumulative shortwave susceptibility of bins
classified as precipitating clouds in Fig.2d, weighted by their frequency of occurrence. The contributions from CF
and $\alpha_c$ effects are $-9.5$ and $-3.5$ $W\ m^{-2}\ ln(N_d)^{-1}$, respectively (Eq. 3).
*b.   Non-precipitating thick clouds*
Non-precipitating thick clouds are less frequent, the total frequency of occurrence is 10% (Fig.2e). Their
cloud LWP responses to $N_d$ perturbations differ from that of precipitating clouds.  The LWP susceptibility for non-
precipitating thick clouds is the most negative among the three cloud states, and it reaches a minimum value of $-1.2$
at the high-LWP and high-$N_d$ ends (Fig.2a). As LWP and $N_d$ decrease, the LWP susceptibility gradually increases
from $-1.2$ to $-0.6$. This negative susceptibility is likely explained by the evaporation enhancement associated with
smaller droplets at high $N_d$ values (e.g., Xue and Feingold, 2006; Small et al., 2009), which works in concert with an
entrainment strengthening expected in clouds with large LWP (e.g., Sandu et al., 2008, Williams and Igel, 2021). In
addition, clouds with higher $N_d$ and larger LWP exhibit stronger shortwave absorption, which enhance LWP
depletion and therefore a more negative LWP susceptibility (e.g. Bores and Mitchell, 1994; Petters et al. 2012). The
mean LWP susceptibility for non-precipitating thick clouds is $-0.94$. Consistent with the negative LWP
susceptibility, non-precipitating thick clouds become less reflective with increasing $N_d$ for all $N_d$ bins with LWP >
75 $gm^{-2}$ (Fig.2b). The mean $\alpha_c$ susceptibility is $-0.04$ $ln(N_d)^{-1}$. Due to the enhanced entrainment and
evaporation, the mean CF mostly decreases with increasing $N_d$, with the mean CF susceptibilities ranging from $-0.1$
to $+0.04\ ln(N_d)^{-1}$ (Fig.2c). Considering the decrease in both $\alpha_c$ and CF, non-precipitating thick clouds exhibit a
warming effect at the surface, the mean radiative response is $+4.4\ W\ m^{-2}\ ln(N_d)^{-1}$(Fig.2d), with contributions
from the albedo effect and the CF effect of 2.9 and 1.5 $W\ m^{-2}\ ln(N_d)^{-1}$, respectively.

*c.   Non-precipitating thin clouds*

Non-precipitating thin clouds are more common than thick clouds during summer, with a total frequency of

occurrence of 44% (Fig.2e). Compared to non-precipitating thick clouds, they exhibit consistent negative but
slightly weaker LWP responses to $N_d$ perturbations. The mean LWP susceptibilities range from $-0.9$ to $-0.4$ in
different LWP-$N_d$ bins with a mean of $-0.7$ (Fig.2a). Similar to non-precipitating thick clouds, non-precipitating
thin clouds mostly become darker with increasing $N_d$. Interestingly, with largely decreased LWP, the mean CF
mostly increase for all $N_d$ conditions, the CF susceptibilities range from $+0.02$ to $+0.25\ ln(N_d)^{-1}$ (Fig.2c).  The
sedimentation- evaporation-entrainment feedback alone cannot explain the opposite signs in in LWP and CF
susceptibilities for non-precipitating thin clouds. A possible explanation for the increased CF is that the enhanced
cloud top radiative cooling rate from aerosol perturbations help to mix the boundary layer, facilitate moisture
transport from the ocean surface to cloud, and therefore favor new cloud formation and extend cloud lifetime (e.g.,
Christensen et al. 2020). This hypothesis is consistent with and supported by the relative low CF for these clouds
(Fig.S2a) and the diurnal variation in LWP susceptibility for non-precipitating thin clouds, which will be discussed
in the next section. The opposite signs of LWP and CF susceptibilities indicate that the AIE might redistribute cloud
water horizontally and make the thin clouds thinner and wider.  The CF radiative effect from increased CF
dominates the albedo effect from darker clouds and lead a net cooling at the surface. The mean radiative response is
$-5.2\ W\ m^{-2}ln(N_d)^{-1}$, with CF and albedo contributions of $-8.3$ and $+3.1\ W\ m^{-2}ln(N_d)^{-1}$, respectively (Fig.2d).

To sum up, the magnitudes and signs of the responses of cloud LWP, $\alpha_c$, and CF to $N_d$ perturbations

primarily depend on the cloud states. Precipitating clouds mostly become thinner and brighter with increasing $N_d$,
accompanied by a slight increase in CF. An increase in LWP with increasing $N_d$ is observed only for heavily
precipitating clouds with $N_d < 30\ cm^{-3}$ and LWP $> 125\ gm^{-2}$. Non-precipitating thick clouds become thinner, less
reflective, and decrease in cloudiness with $N_d$ perturbations. On the other hand, non-precipitating thin clouds
become thinner and less reflective, but their cloudiness increase as $N_d$ increases. Given the dependence of AIE on
cloud state, we will apply the cloud state classification in the following two sections with the goal of facilitating a
process-level understanding of cloud responses and the daytime variation in cloud susceptibilities.

**3.3 Daytime variation of cloud susceptibility**

As discussed in the introduction, warm boundary layer clouds exhibit a distinct diurnal cycle in both cloud

properties and frequency of occurrence of cloud states during summer. In this section, we investigate the daytime
variation of cloud susceptibility from 9 to 18 local standard time (LST) using the half-hourly Meteosat-11 retrievals.
The domain mean daytime variation of cloud susceptibility is estimated from each half-hourly time step within each
$1° \times 1°$ box and then averaged over the study domain (33-43°N, 23-33°W) during the four months. In the study
domain, there is little spatial variability in cloud susceptibilities and the diurnal cycle of the cloud susceptibility for
the $1° \times 1°$ box at the ARM ENA site agree well with the domain mean pattern (not shown). Furthermore, diurnal
cycle of the cloud microphysical properties (e.g., $r_e$, $\tau$, LWP, $N_d$) show little difference between the domain mean
value or that averaged over the $1° \times 1°$ box at the ARM ENA site. The cloud microphysics retrievals from
Meteosat-11 agree well with retrievals based on ground-based radar and lidar observations in the daytime variation
(not shown). Therefore, the ARM ENA site at the Azores archipelago can represent the cloud properties and the AIE
for warm boundary layer clouds over the study region.

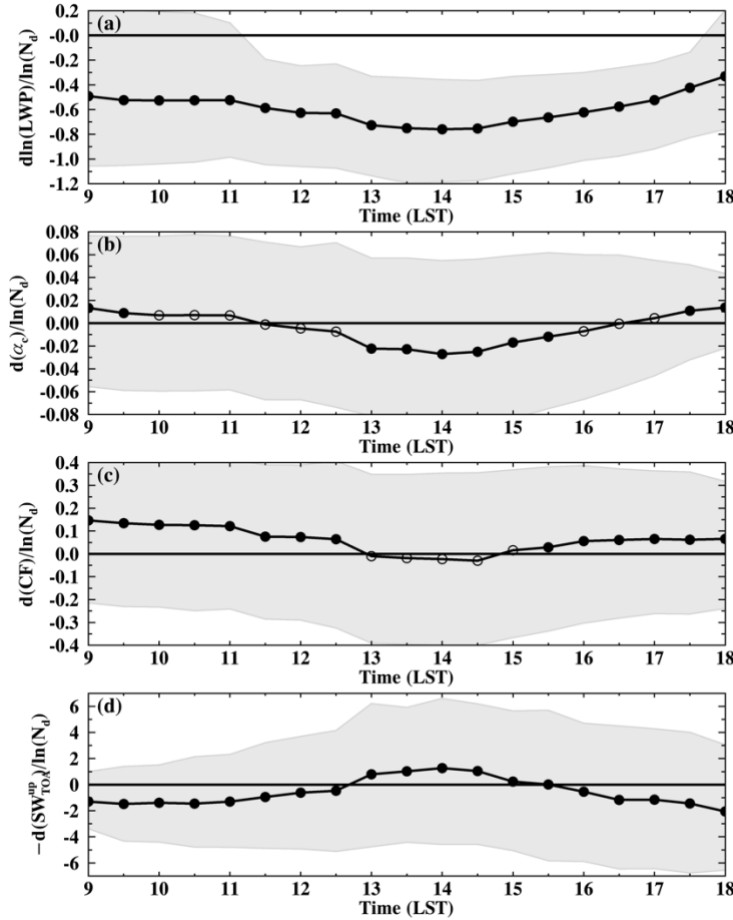

Figure 3. Daytime variation of cloud susceptibilities. (a) cloud LWP susceptibility ($dln(LWP)/dln(N_d)$), (b) cloud
albedo susceptibility ($d\alpha_c/dln(N_d)$), (c) cloud fraction susceptibility ($dCF/dln(N_d)$), and (d) cloud shortwave
susceptibility ($-dSW_{TOA}^{up}/dln(N_d)$). The shaded areas represent the lower and upper 25th percentile of the cloud
susceptibilities for each time step and the solid lines with dots represent the mean values. In (b) and (c), filled
markers indicate data points that susceptibilities are significantly different from zero (p<0.05), while open markers
indicate statistical insignificance.

Warm boundary layer clouds witness distinct and significant daytime variations in cloud susceptibilities
(Fig.3). For example, the mean LWP susceptibility exhibits a magnitude of change of 0.4 from morning to evening,
which corresponds to approximately 30-40% of the overall variability in LWP susceptibility (Fig.3a). Similarly, the
$\alpha_c$ and CF susceptibility undergo magnitude of changes of approximately 20-30% compared to the overall
variability (Figs. 3b and c). The high variability in cloud susceptibility highlights the complex synoptic,
meteorological and cloud conditions as well as the interplay between them in the ENA region. Nevertheless, the
daytime variation of cloud susceptibility is statistically significant at a 95% confidence level based on a student's t-
test. Interestingly, all three cloud variables exhibit a "U-shaped" diurnal cycle in cloud susceptibilities with less
negative/more positive values in the morning and evening and more negative values at noon. Additionally, both $\alpha_c$
and CF susceptibilities switch signs from positive in the morning to negative at noon, and then become positive
again in the evening. The switch in sign for albedo susceptibility is statistically significant at a 95% confidence
level, while the switch in sign for CF susceptibility is not statistically significant (Figs. 3b, c). As both $\alpha_c$ and CF
increase with increasing $N_d$ in the morning, AIE has a cooling effect at the surface and the estimated shortwave
susceptibility is $-1.4\ W\ m^{-2}\ ln(N_d)^{-1}$. During 13-15 LST, the shortwave susceptibility switches sign to a warming
effect of $+1.2\ W\ m^{-2}\ ln(N_d)^{-1}$ (Fig.3d).

Given the pronounced daytime variation of cloud susceptibility, *how can we explain this distinct daytime*

*variation, and which state of cloud contributes most to the daytime variation?* One possible explanation is the
increased occurrence of precipitating clouds in the morning and evening during summer (Remillard et al, 2012),
which increase cloud susceptibility, as depicted in Fig.2. To investigate this hypothesis and quantify the impacts of
different cloud states on the variabilities of cloud susceptibilities, we examined the daytime variation of cloud
susceptibility, along with the daytime shift in occurrence frequency for each cloud state.
**3.4 Daytime variation of cloud susceptibility for different cloud states**
**3.4.1 Non-precipitating thin clouds**

Non-precipitating clouds mainly consist of thin clouds, with a daytime mean occurrence of 44% (Fig.4a).

The highest occurrence is observed around noon, which is consistent with ground-based radar reflectivity
measurement at the ENA site (Remillard et al, 2012). Furthermore, as seen in Fig.4, not only the frequency of cloud
occurrence, but also the susceptibilities of LWP, $\alpha_c$, and CF show distinct daytime fluctuations. For example, the
mean LWP susceptibility decreases from $-0.4$ to $-0.9$, and the mean $\alpha_c$ susceptibility decreases from 0.02 to $-0.04$
$ln(N_d)^{-1}$ from morning to noon, followed by increases in both LWP and $\alpha_c$ susceptibilities in the afternoon. The
CF susceptibility is highly positive in the morning and decreases to near zero after 13 LST. In addition, cloud
susceptibility for thin clouds in the morning is statistically significantly different from that at noon and in the
evening at a 95% confidence level.

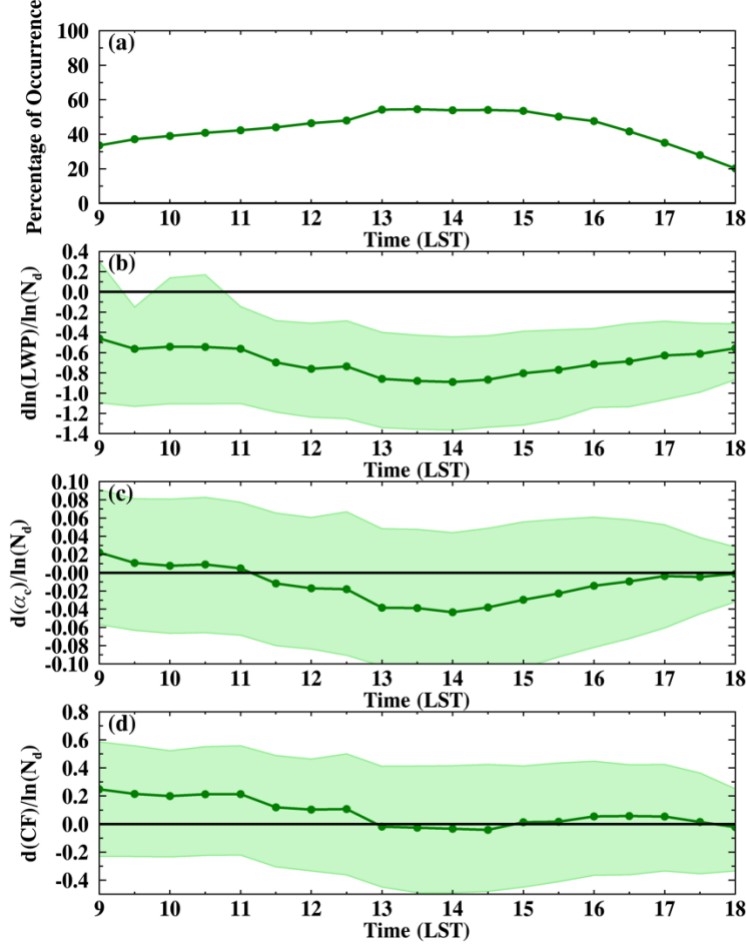

Figure 4. Daytime variation of (a) percentage of occurrence of non-precipitating thin clouds to warm boundary layer clouds, (b) cloud LWP susceptibility ($dln(LWP)/dln(N_d)$), (c) cloud albedo susceptibility ($d\alpha_c/dln(N_d)$), and (d) cloud fraction susceptibility ($dCF/dln(N_d)$) for non-precipitating thin clouds. The shaded areas represent the lower and upper 25[th] percentile of the cloud susceptibilities for each time step and the solid lines with dots represent the mean values.

To explain the decrease of cloud susceptibility of non-precipitating thin clouds from morning to noon, we test two hypotheses (H1 and H2 in Table 1). Hypothesis H2 is related to the dissipation of thin clouds during this period, which is caused by a decreased LWP due to increased solar radiation. During the dissipation, both LWP and $r_e$ decrease. As $r_e$ is raised to the power of $-\frac{5}{2}$ in Eq. (1) compared to $\tau$ being raised only to the power of $\frac{1}{2}$, the decreases of LWP and $r_e$ could still result in an increase in the retrieved $N_d$. Consequently, a LWP decrease and $N_d$ increase lead to a decrease in LWP susceptibility during the dissipation (Gryspeerdt et al., 2019). To examine this hypothesis (H2), non-precipitating thin clouds are classified as: growing, dissipating, or constant based on the changes in the mean CF, cloud susceptibilities for the three groups are shown in Fig.S3. More specifically, we calculate the change in the mean CF within a 30-minute window for each fixed $1° \times 1°$ box. If the mean CF increase (decrease) more than 10%, clouds are classified as growing (dissipating). If the change in CF is less than 10%, clouds are classified as constant. Similar results are obtained using classification methods based on different CF thresholds (e.g., from 10% to 30%), and during different time windows from 30-minutes to two hours (not shown).

Table 1. List of hypotheses and associated explanations for the daytime variation of LWP and CF susceptibilities for different cloud states.

| | Cloud state | Hypotheses: | |
|---|---|---|---|
| Daytime evolution of LWP susceptibility | Non-precipitating thin clouds | H1*: | Non-precipitating thick clouds transition to thin clouds from morning to noon, which leads to a daily minimum LWP susceptibility at noon. |
| | | H2*: | Cannot explain. Clouds that are growing or dissipating have similar LWP susceptibilities as clouds with constant CF. |
| | Non-precipitating thick clouds | H1: | Thin clouds develop to thick clouds from noon to evening, which leads to an increase in LWP susceptibility. |
| | | H2: | Cannot explain. |
| | Precipitating clouds | H1: | Non-precipitating thin clouds transition to precipitating clouds in the afternoon, which leads to a decrease in LWP susceptibility. |
| | | H2: | Cannot explain. |
| Daytime evolution of CF susceptibility | Non-precipitating thin clouds | H1: | Thick clouds transitioned to thin clouds from morning to noon, leading to a decrease in CF susceptibility |
| | | H2: | Cannot explain. |
| | Non-precipitating thick clouds | H1: | Cannot explain. |
| | | H2: | Cannot explain. |
| | | H3*: | Mostly overcast clouds in the morning and evening. CF of overcast clouds is less sensitive to $N_d$ perturbations. |
| | Precipitating clouds | H1: | Thin clouds transition to precipitating clouds in the afternoon, and lead to a decrease in CF susceptibility |
| | | H2: | Cannot explain. |

*H1: LWP and CF responses to $N_d$ perturbations slower than the transition of cloud state.
*H2: Dissipation or development of clouds.
*H3: Changes in cloud morphology.

As seen in Fig. S3b, the LWP susceptibilities for non-precipitating thin clouds in the growing or dissipating stages are similar or less negative than clouds that remain constant in CF, which contradicts the hypothesis H2. Additionally, the occurrence of dissipating and developing thin clouds remain relatively constant throughout the day (Fig.S3a), which differs from our hypothesis that thin clouds dissipate in the morning. Therefore, the decrease in LWP susceptibility in the morning is *unlikely* to be attributed to the dissipation or development of thin clouds. Yet, due to the observational limitation on estimating the mixing process from satellite retrievals, further investigation is needed to quantify the impact of cloud dissipation on the $N_d$-LWP relationship.

Besides the change in CF, dissipation/development of clouds can be defined by change in LWP. However, as our definition of thin and thick clouds use LWP thresholds, results based on change in LWP are similar to results shown in Fig.5, but with weaker signal (not shown). This indicates that classification of precipitating versus non-precipitating clouds is necessary in distinguishing cloud responses to $N_d$ perturbations rather than merely using the LWP threshold.

Hypothesis H1 is related to the response time of cloud LWP and CF to $N_d$ perturbations. Both numerical
models and observations have shown that the influence of aerosols on cloud LWP, achieved through adjusting the
entrainment rate, may take four hours to become apparent and up to 20 hours to reach an equilibrium (e.g.,
Glassmeier et al. 2021; Gryspeerdt et al., 2021; Fons et al., 2023). Similarly, CF increases gradually from increasing
aerosols and may take approximately three to four hours to reach its maximum effect after the initial perturbation
(Gryspeerdt et al., 2021). Therefore, we hypothesize that if clouds change state during the adjustment time, clouds
may still retain the "memory" of their responses to $N_d$ perturbations from the previous state. The possible physical
processes and mechanisms for this hypothesis is that the LWP susceptibility is mainly driven by cloud top
evaporation and entrainment rate. The positive feedback among entrainment, evaporative cooling, long-wave
radiative cooling, and mixing from cloud top form a positive feedback loop and set up an environment conductive to
enhanced entrainment and evaporation. These feedback and environment will not change immediately even when
the cloud LWP decrease and cloud transition to a thin state or vice versa. With the diurnal variation in cloud
properties and transition in cloud state, it leads to a diurnal evolution in cloud susceptibility.
To quantify the dependence of current cloud susceptibility on previous cloud states, we track the cloud state
for each $1° \times 1°$ box backward in time for two hours and classify the non-precipitating thin clouds into three groups
(Fig.5): (1) thin clouds that are currently classified as thin clouds and didn't change states in the past two hours (thin
→ thin), (2) thin clouds that evolved from precipitating clouds (rain → thin), and (3) thin clouds that decayed from
non-precipitating thick clouds (thick → thin). This backward tracking classification is applied at each time step. As
shown in Fig.5a, at 9 LST, ~50% of the non-precipitating thin clouds originate from thick clouds in previous hours.
The transition from thick to thin clouds is likely caused by the increased solar radiation after sunrise, leading to
cloud turbulent decoupling from the ocean surface and a decrease in cloud LWP. In the evening, on the other hand,
around 80% of the thin clouds are thin clouds in previous hours. In addition, less than 20% of the non-precipitating
thin clouds transition from precipitating clouds.
In consistent with our hypothesis, non-precipitating thin clouds that are previously thick have significantly
more negative LWP and $\alpha_c$ susceptibilities than thin clouds that are previously thin or precipitating (Figs. 5b and c).
The differences between the two categories are most pronounced from late morning to early afternoon and less
pronounced in the early morning and evening. Such pattern is likely attributed to the daytime evolution of marine
boundary layer and cloud coupling state. For example, in the early morning (e.g., 9-10 LST), even with higher
frequency of occurrence of thick clouds transitioning to thin clouds, the LWP susceptibility for the thick-to-thin
category is less negative compared to later time (dashed line with diamond symbols in Figs 5a, b). This is attributed
to the less negative LWP susceptibility for non-precipitating thick clouds in earlier time (e.g., 7-9 LST, not shown),
in connection with a well-mixed boundary layer able to transport moisture from the ocean to the cloud, which
compensates the moisture loss from aerosol-enhanced entrainment (e.g., Sandu et al., 2008), so that both thick and
thin clouds exhibit less negative LWP susceptibilities. From late morning to early afternoon, with increasing solar
radiation, deepening of boundary layer and clouds decoupled from surface, LWP susceptibility for thick clouds
largely decreases and reaches a daily minimum, which contributes to the largest difference between the thin-to-thin
and thick-to-thin categories shown in Fig.5b. The opposite processes occur from afternoon to evening, LWP of thick
clouds become less susceptible to $N_d$ perturbations, and the difference between the two categories is less
pronounced. These results support our hypothesis that clouds retain the memory of their responses to $N_d$
perturbations from their previous states.

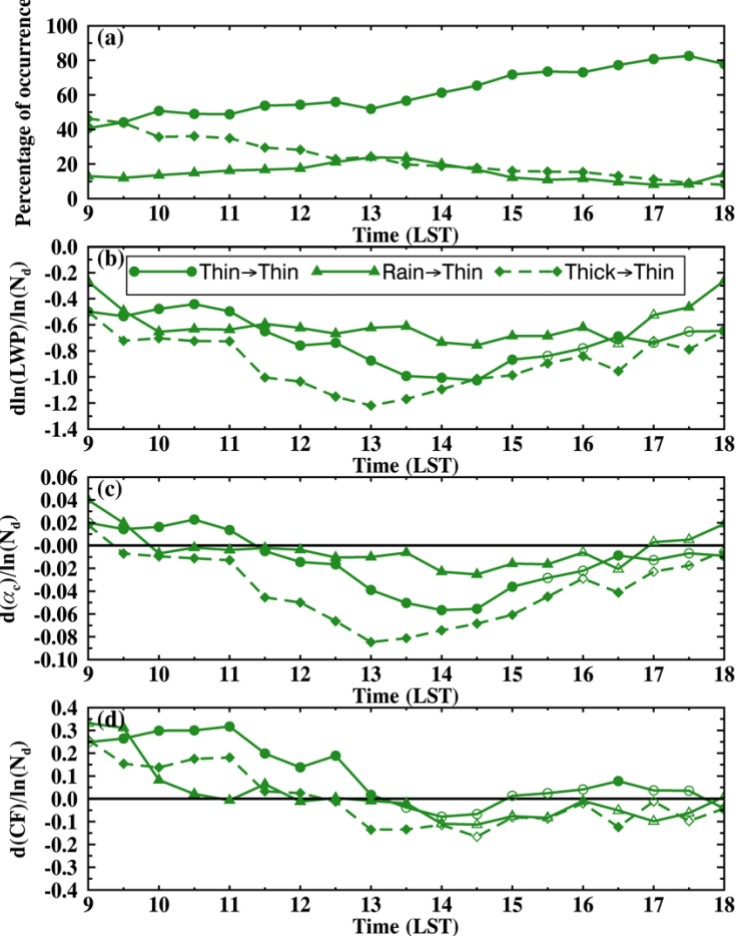


Figure 5. Daytime variation of non-precipitating thin clouds transition from non-precipitating thin clouds (thin →
thin, solid line with circle symbols), precipitating clouds (rain → thin, solid line with triangle symbols), and non-
precipitating thick clouds (thick → thin, dashed line with diamond symbols) in previous two hours. Symbols for
different state transitions are noted in (b). In (b)-(d), filled markers indicate data points that are significantly
different from the other two groups (p<0.05), while open markers indicate statistical insignificance.

Similar to LWP, responses of CF to $N_d$ perturbations in the morning retain the memory of the previous

state of clouds. As seen in Figure 5d, thin clouds that transitioned from thick clouds or precipitating clouds have
significantly less positive CF susceptibility than thin clouds that were previously thin, particularly in the morning.
As the CF susceptibility for thin clouds that evolved from precipitating and thick clouds greatly decrease from
morning to noon, the CF susceptibility for thin clouds decrease from large positive to near zero from morning to
noon (Fig.4c). A maximum in CF susceptibility in the early morning is likely associated with the influence of
aerosols on boundary layer mixing and the evolution of boundary layer from morning to noon. The enhanced
entrainment rate and radiative cooling rate from $N_d$ perturbations help to destabilize the boundary layer and
transport moisture from the ocean surface to clouds, facilitating new cloud formation (e.g., Christensen et al. 2020).
As the boundary layer is typically well mixed in the morning with clouds coupled to the surface, the impact of
aerosols on CF is strongest in the morning and gradually decrease from morning to noon. In the afternoon, thin
clouds transition from all three states have near-zero CF responses to $N_d$ perturbations. Further analyses and model
simulations are needed to better understand the impacts of aerosols and the associated diurnal evolution of
entrainment rate, boundary layer mixing on cloud cover and lifetime, to better explain the observed daytime
variation of CF susceptibility for non-precipitating thin clouds.

Lastly, the impact of the cloud memory of AIE on current cloud susceptibility is also evident within a 30-

minute window when a transition of cloud state just occurs (Fig.S4). Consistent with the findings in Fig.5, thin
clouds that transition from thick clouds exhibit much more negative LWP and $\alpha_c$ susceptibilities compared to thin
clouds that remain thin during the 30 minutes. Yet, the number of cases experiencing a transition in cloud state
within a 30-miniute window is limited (Fig.S4a). In addition, the impact of the transition in cloud state on the
current cloud susceptibility persists for at least four hours (Fig.S5). As our tracking method does not follow
individual cloud parcels to track changes in their states, the influence of cloud advection may become significant
over longer tracking time, such as four hours. Therefore, a two-hour tracking window is used in this study.

As discussed in the method section, while the advective effects in our study are expected to be modest, we

further isolate their impact, by performing an analysis for cloud scenes with wind speed of less than 7 m/s (60% of
time), when clouds are somewhat stationary in two hours. Influence of transition in cloud state is consistent as in
Fig.5 with more negative LWP and $\alpha_c$ susceptibilities for thin clouds transitioned from thick clouds, while the signal
is slightly stronger (not shown). This consistency confirms that our tracking method can capture the signal of cloud
state transition and its impact on cloud susceptibilities during summer in the study region.

In summary, the "U-shaped" daytime variations in LWP and $\alpha_c$ susceptibilities for non-precipitating thin

clouds are likely due to cloud retaining the memory of AIE. From morning to noon, as non-precipitating thick
clouds evolve to thin clouds, they retain their memory of the large negative LWP susceptibility. Therefore, both
LWP and $\alpha_c$ susceptibilities decrease from morning to noon for thin clouds and reach their daily minima at noon. In
the afternoon, as a growing percentage of thin clouds persist as thin clouds during the next hours, LWP and $\alpha_c$
susceptibilities gradually increase to less negative and near zero, respectively.

**3.4.2 Non-precipitating thick clouds**

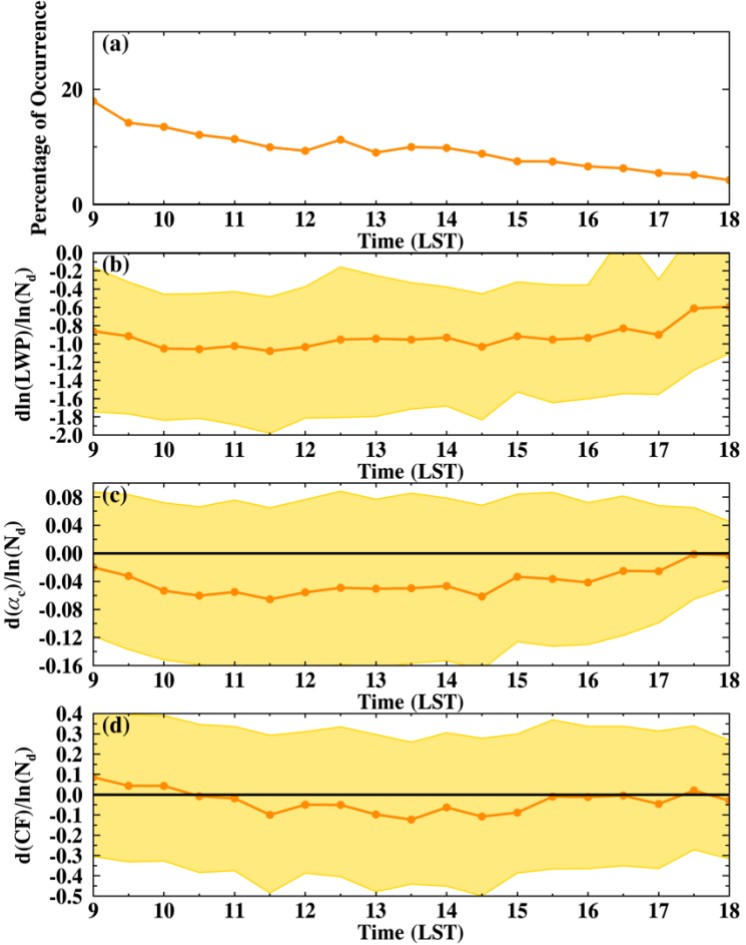

Figure 6. Daytime variation of (a) percentage of occurrence of non-precipitating thick clouds to warm boundary layer clouds, (b) cloud LWP susceptibility ($dln(LWP)/dln(N_d)$), (c) cloud albedo susceptibility ($d\alpha_c/dln(N_d)$), and (d) cloud fraction susceptibility ($dCF/dln(N_d)$) for non-precipitating thick clouds. The shaded areas represent the lower and upper 25th percentile of the cloud susceptibilities for each time step and the solid lines with dots represent the mean values.

Consistent with Fig.2e, non-precipitating thick clouds are the least frequent warm boundary layer cloud state during summer over the ENA region. Their percentage of occurrence continuously decreases from 20% in the morning to less than 5% in the evening (Fig.6a). For LWP and $\alpha_c$, their susceptibilities first decrease from less negative to more negative in the morning and then increase from noon to evening (Fig.6b and c, respectively). CF susceptibility is weakly positive in the early morning, becomes weakly negative from late morning to early afternoon, and increases to near zero in the evening (Fig.6d). The daytime evolutions of LWP and $\alpha_c$ susceptibilities for thick clouds exhibit consistent trend with cloud susceptibilities for thin clouds transition from thick clouds shown in Fig.5 but with a lag of two hours. For example, the LWP susceptibility for thick clouds decreases from $-0.8$ to $-1.1$ from 9 to 11 LST and it increase from $-1.1$ to $-0.8$ from 11 to 16 LST; while the LWP susceptibility for the thick-to-thin category in Fig.5b decreases from $-0.8$ to $-1.2$ from 11-13 LST and increases to $-0.6$ from 13 to 18 LST. This result supports our hypothesis on cloud retaining its memory of AIE of its previous cloud state.

To gain insight into the observed evolution of LWP and $\alpha_c$ susceptibility from morning to evening, we

investigate the influence of cloud state transition on cloud susceptibility for non-precipitating thick clouds, which is
summarized as H1 in Table 1. As shown in Fig.7a, around 40% of thick clouds sustain as thick clouds from the
previous two hours in the morning; whereases during the late afternoon to evening, with decreasing solar radiation,
more than 60% of thick clouds are developed from thin clouds. Consistent with the findings presented in Fig.5, thick
clouds that are previously thick exhibit significantly more negative LWP susceptibility compared to thick clouds that
are previously thin (Fig.7b). These differences are particularly prominent from late morning to noon and become
insignificant in the afternoon. As discussed before, difference between the thick-to-thick and the thin-to-thick
categories are due to the LWP susceptibilities for thick and thin clouds of previous time, while the smaller
differences in the early morning and afternoon could be attributed to the expected stronger turbulence and cloud
coupling at these times. Additionally, Fig.7d indicates that transition in cloud state cannot explain the daytime
variation in CF susceptibility for thick clouds, as all three groups are insignificantly different from each other.

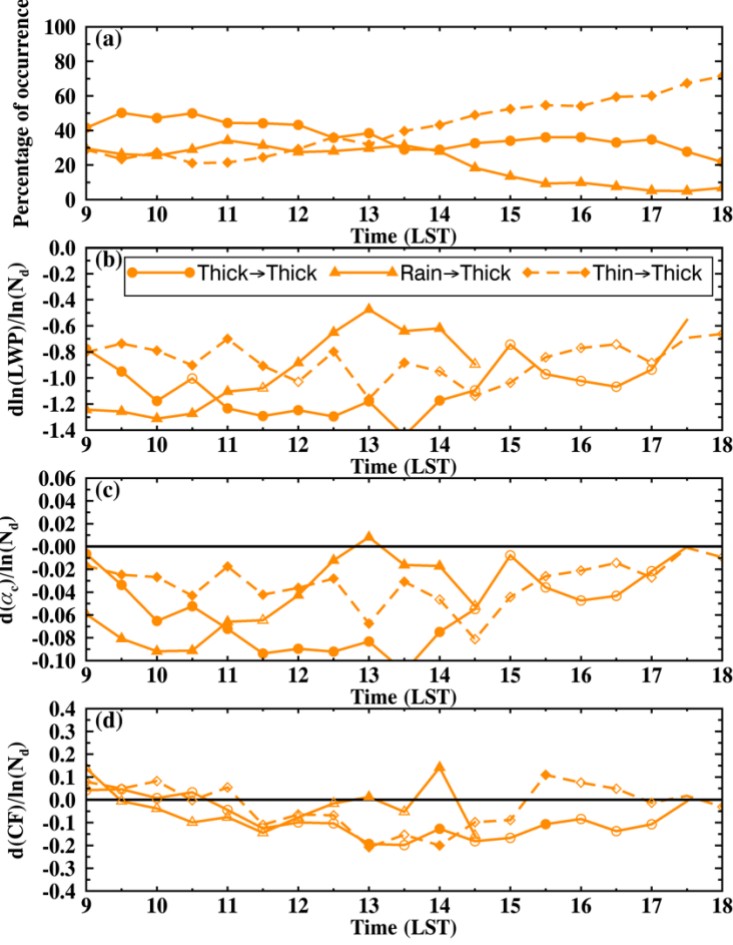

Figure 7. Daytime variation of non-precipitating thick clouds transition from non-precipitating thick clouds (thick →
thick, solid line with circle symbols), precipitating clouds (rain → thick, solid line with triangle symbols), and non-
precipitating thin clouds (thin → thick, dash line with diamond symbols) in previous two hours. Symbols for
different state transitions are noted in (b). In (b)-(d), filled markers indicate data points that are significantly
different from the other two groups (p<0.05), while open markers indicate statistical insignificance.

To understand the driving mechanism for the daytime variation in CF susceptibility shown in Figure 6d, we

calculate the mean cloud properties for non-precipitating thin and thick clouds, as shown in Fig.8. In the morning,
non-precipitating thick clouds are predominantly overcast clouds with a mean CF of 75% (Fig.8a). To distinguish
between overcast and broken clouds, we calculate the diameter-to-height ratio (DHR) for each cloud, where
diameter is estimated by the square root of the area and height is defined as the 90th percentile of cloud tops. As
shown in Fig.8c, thick clouds are mostly overcast in the morning with a mean DHR of 230. Compared to broken
clouds, overcast clouds have less room for CF to increase, which results in a less positive CF susceptibility for thick
clouds compare to thin. After 10 am, non-precipitating thick clouds start to break. The mean CF decreases from 75%
at 10 am to 60% at 2 pm and the DHR decreases from 230 to 170. As CF for broken clouds is more sensitive to $N_d$
perturbations, CF susceptibility decreases to $-0.13\ ln(N_d)^{-1}$, which is consistent with the daytime mean negative
CF susceptibility shown in Fig.2c. From afternoon to evening, clouds transition to overcast again (Fig.8), and the CF
susceptibility increases back to zero. This impact of cloud morphology (e.g., overcast or broken clouds) on daytime
variation of CF susceptibility is summarized as H3 in Table 1.

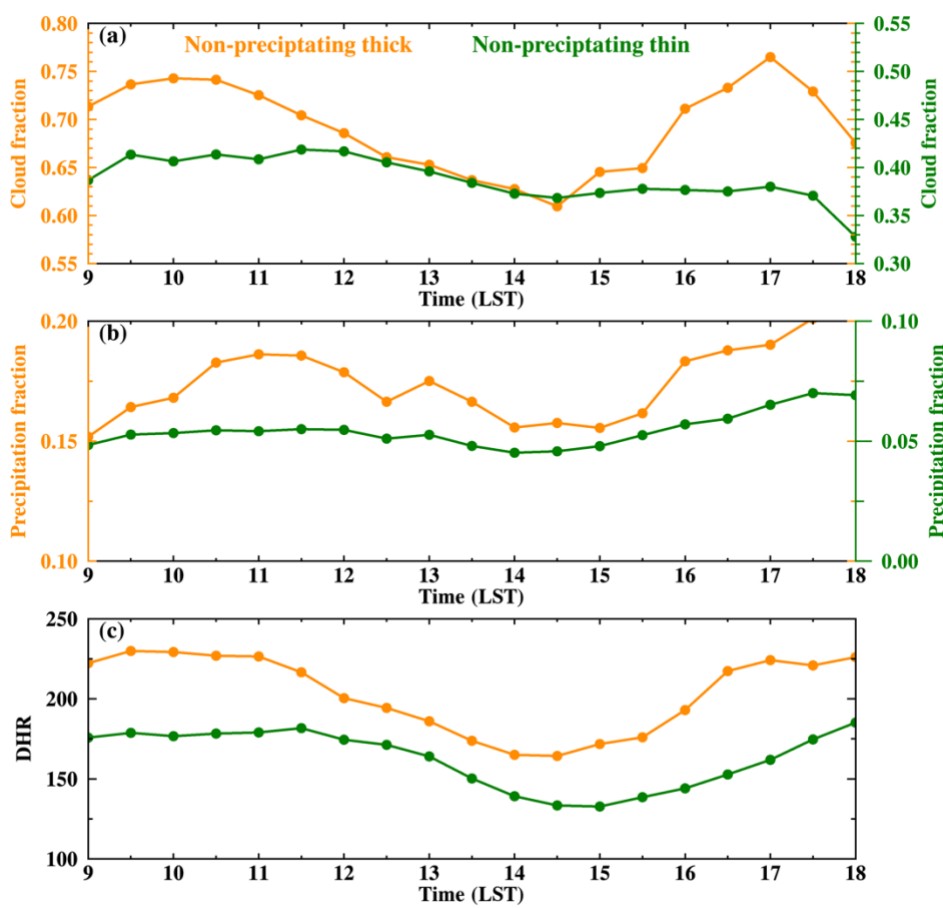

Figure 8. Diurnal variation of (a) cloud fraction, (b) pixel-level precipitation fraction, and (c) diameter-to-height
ratio (DHR) for non-precipitating clouds. Different colors represent different cloud states as indicated in (a). Please
note that the non-precipitating thin cloud in (a) and (b) use the y-axis on the right side.
The previous results are summarized as follows: LWP susceptibility for non-precipitating thick clouds first
decreases from less negative to more negative in the morning and then increase from noon to evening, which is
likely attributed to the transition from thin to thick clouds. In the morning, 40% to 50% of thick clouds are
previously thick clouds, these clouds exhibit a large negative LWP susceptibility. In the afternoon, with increasing
percentage of thick clouds develop from thin clouds and retain the memory of LWP responses to $N_d$ perturbations of
the thin clouds. LWP susceptibility gradually increases and becomes similar to that of thin clouds (Fig.4b, 6b).
Daytime variation in CF susceptibility for thick clouds is likely attributed to changes in cloud morphology. In the
morning and evening, thick clouds are mostly overcast with CF less sensitive to $N_d$ perturbations, resulting in a near
zero CF susceptibility. From late morning to early afternoon, the overcast thick clouds break down and CF decrease
with increasing $N_d$ likely due to the increased shortwave absorption, the enhanced entrainment, and evaporation.
The impact of cloud memory and transition of cloud state on the daytime variation of LWP susceptibility is
summarized as a schematic figure shown in Fig.9. From morning to noon, as non-precipitating thick clouds
transition to thin clouds, thin clouds retain their memory of AIE of their previous state. Therefore, LWP
susceptibility for thin clouds decreases from morning to noon and reach its daily minima in the early afternoon.
From early afternoon to evening, with non-precipitating thin clouds developing to thick clouds, LWP susceptibility
for thick clouds increase.

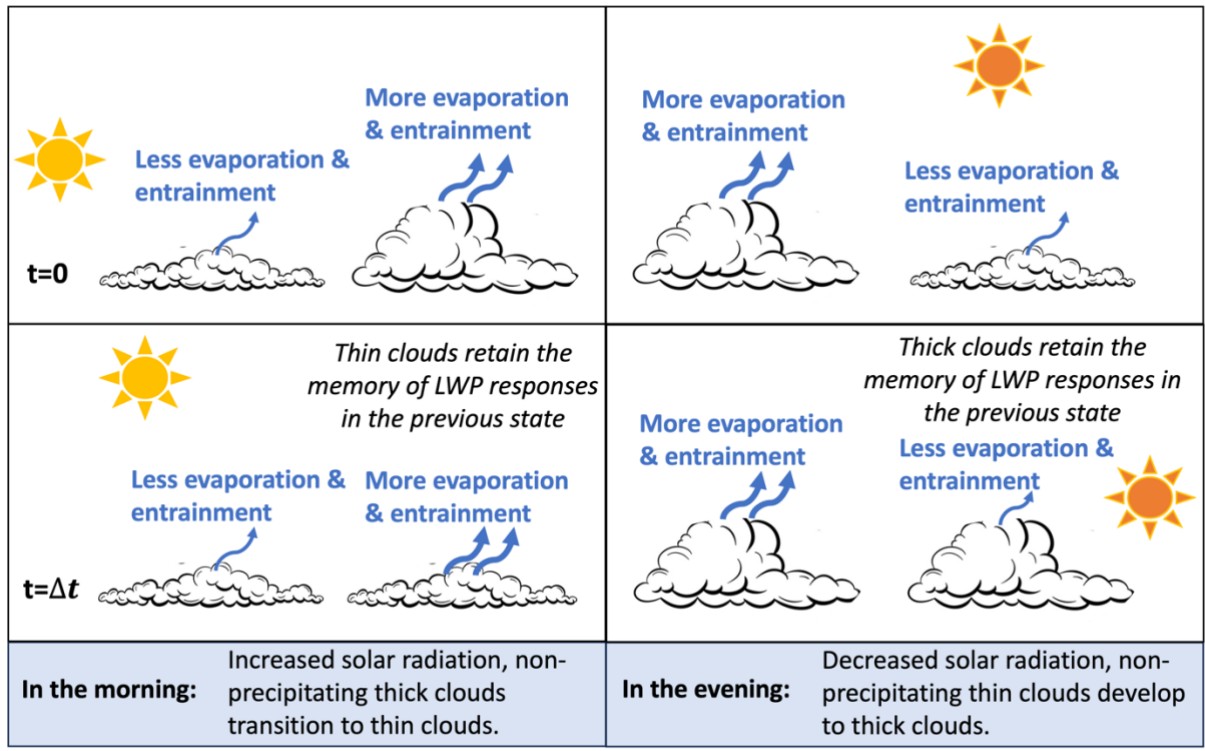

Figure 9. Schematic figure of influence of cloud memory and transition of cloud state on the LWP susceptibility and
its daytime variation.

**3.4.3 Precipitating clouds**

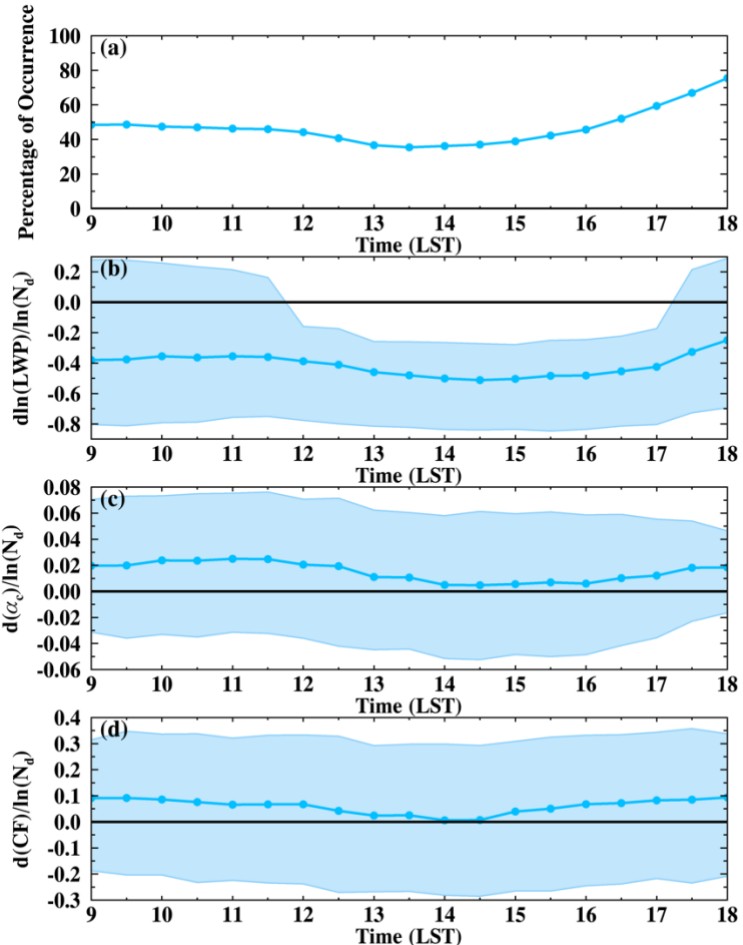

Figure 10. Daytime variation of (a) percentage of occurrence of precipitating clouds to warm boundary layer clouds, (b) cloud LWP susceptibility ($dln(LWP)/dln(N_d)$), (c) cloud albedo susceptibility ($d\alpha_c/dln(N_d)$), and (d) cloud fraction susceptibility ($dCF/dln(N_d)$) for precipitating clouds. The shaded areas represent the lower and upper 25[th] percentile of the cloud susceptibilities for each time step.

Precipitating clouds, depicted in Figs.10a, are the dominant cloud state in this region, accounting for 46% of the warm boundary layer clouds, compared to 44% of non-precipitating thin clouds. The frequency of precipitating clouds is higher in the morning and evening compared to noon. Throughout the day, the mean LWP susceptibility remain consistently negative, fluctuating between −0.5 to −0.3, with minimum values between 14–16 LST (Figs.10b). The daytime variability in LWP susceptibility for precipitating clouds is much lower than that for non-precipitating thin (e.g., from −0.9 to −0.4) and thick (e.g., from −1.1 to −0.6) clouds. The negative LWP susceptibility is likely due to the prevalence of lightly precipitating clouds, with a mean precipitating fraction ranging from 0.2 to 0.5 (Fig.S2d). The influence of precipitation suppression is smaller than that of the entrainment enhancement. Similarly, $\alpha_c$ susceptibility fluctuates between 0 to 0.02 throughout the day, with near zero $\alpha_c$ susceptibility in early afternoon (Figs.10c). Despite the minimal daytime variation, the LWP and $\alpha_c$ susceptibilities at 13-16 LST are statistically significant different from cloud susceptibilities in the morning and evening at 95%

confidence level with the two-tailed t-test. The CF susceptibility for precipitating clouds also shows minimal
daytime variation compared to non-precipitating clouds, with a mean value ranging from 0 to 0.1 (Figs.10d).

Consistent with non-precipitating clouds, the daytime variation of LWP and $\alpha_c$ susceptibilities for

precipitating clouds can be attributed to the transition of cloud states. For example, as shown in Figs.11b-d,
precipitating clouds that transition from non-precipitating thin clouds exhibit significantly more negative/less
positive cloud susceptibilities than precipitating clouds that are previously precipitating. Meanwhile, $\alpha_c$ and CF
susceptibilities switch signs from positive to negative in the afternoon for precipitating clouds transition from non-
precipitating thin clouds compared to that are previously precipitating (dash line with diamond symbols in Figs. 11c,
d). Starting from 13 LST, when non-precipitating thin clouds transition to precipitating clouds (Fig.11a), LWP and
$\alpha_c$ susceptibilities begin to decrease and reach their daily minimum in the late afternoon. Interestingly, as non-
precipitating clouds transition to precipitating clouds (Figs.11b and c, thin → rain, thick → rain), their LWP and $\alpha_c$
susceptibilities exhibit both less negative values and smaller daytime variations compared to thin/thick clouds that
remain as thin/thick (Figs. 5b and c, thin → thin, Figs. 7b and c, thick → thick). The underlying reason for this
observation is currently unclear and warrants further investigation on the sensitivity of AIE for clouds experiencing
transition in cloud states, especially between precipitating and non-precipitating clouds. Lastly, the percentage of
precipitating clouds that transition from non-precipitating thick clouds is less than 7% (Fig.11a).

In conclusion, precipitating clouds exhibit smaller daytime variation in cloud susceptibilities compared to

non-precipitating thin and thick clouds. The decrease of LWP and $\alpha_c$ susceptibilities for precipitating clouds in the
afternoon is likely contributed by the transition of non-precipitating thin clouds to precipitating clouds.

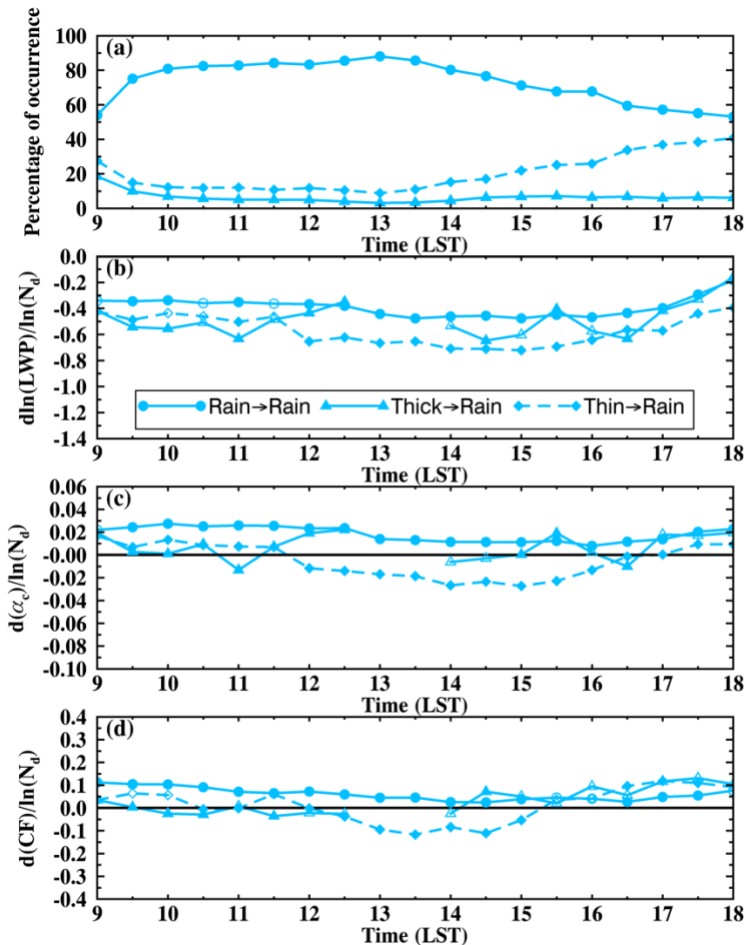

Figure 11. Daytime variation of precipitating clouds transitioned from precipitating clouds (rain → rain, solid line with circle symbols), non-precipitating thick clouds (thick → rain, solid line with triangle symbols), and non-precipitating thin clouds (thin → rain, dash line with diamond symbols) in previous two hours. Symbols for different state transitions are noted in (b). In (b)-(d), filled markers indicate data points that are significantly different from the other two groups (p<0.05), while open markers indicate statistical insignificance.

Combining the results shown here and results in section 3.4.1, we can answer which cloud state contributes the most to the daytime variation of cloud susceptibility. The non-precipitating thin clouds exhibit similar daytime variations in LWP, $\alpha_c$, and CF susceptibility than the warm boundary layer clouds (Fig.4 vs. Fig.3), with clouds being less susceptible to $N_d$ perturbations in the morning and evening and more susceptible at noon. Additionally, non-precipitating thin clouds have highest frequency at noon. On the other hand, precipitating clouds, despite their higher percentage of occurrence than thin clouds, exhibit minimal daytime variation in cloud susceptibility. Therefore, the pronounced daytime variations in cloud susceptibilities for warm boundary layer clouds primarily stem from non-precipitating thin clouds. The distinct daytime evolution patterns for the three clouds states highlight the importance of cloud state classification in quantification of cloud susceptibility.

**3.5 Contribution to the daytime variation of cloud susceptibility**

As discussed in the previous section, both the frequency of occurrence of cloud states and the intensity of cloud responses to $N_d$ perturbations exhibit pronounced daytime variations. In this section, we aim to compare the

contribution of these two components to the overall daytime variation in cloud susceptibilities by fixing one
component constant at a time. The contribution from changes in the frequency of cloud states is represented by the
red lines in Fig.12, which is estimated by weighting the daytime mean cloud susceptibility (Figs. 2a-c) with the half-
hourly frequency of occurrence of clouds in the LWP-$N_d$ parameter space, assuming a constant intensity of cloud
susceptibility during the daytime. The contribution from changes in the intensity of cloud susceptibility is depicted
by the blue lines, which is estimated by weighting the half-hourly cloud susceptibility in the LWP-$N_d$ parameter
space with the daytime mean frequency of occurrence of clouds (Fig.2e), assuming a constant frequency during the
daytime. The black line in Fig.12 represents the observed susceptibility shown in Fig.3, and it includes the
contributions from daytime variations in both components.

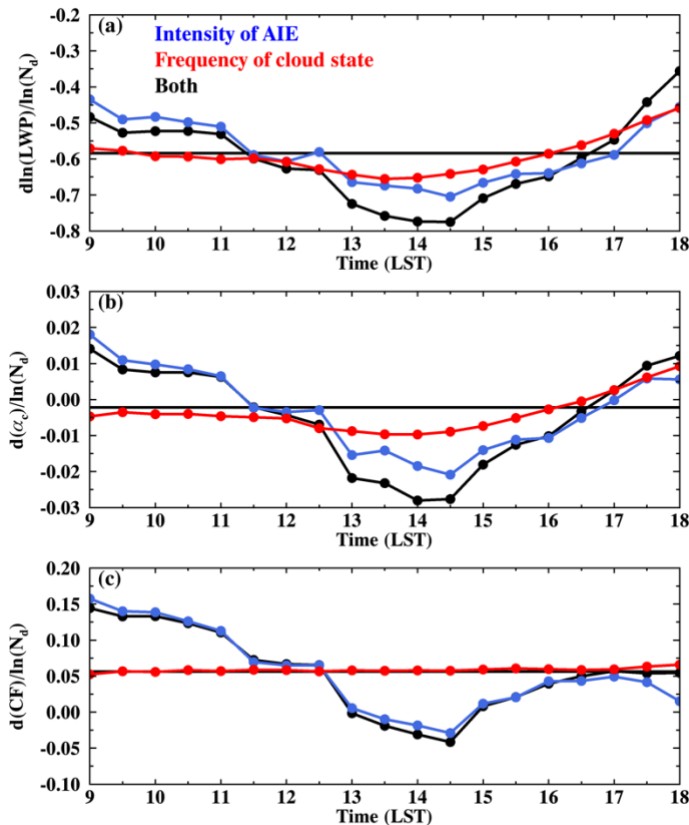

Figure 12. Daytime variation in cloud susceptibility contributed from the variability in the intensity of susceptibility
(blue lines with symbols), variability in the frequency of occurrence of cloud state (red lines with symbols), and
from both (black lines with symbols). (a) cloud LWP susceptibility ($dln(LWP)/dln(N_d)$), (b) cloud albedo
susceptibility ($d\alpha_c/dln(N_d)$), (c) cloud fraction susceptibility ($dCF/dln(N_d)$). The horizontal black solid lines in
(a)-(c) are the daytime mean susceptibility.

When comparing the net observed daytime variation of cloud susceptibilities (black lines) with the

contributions from changes in the intensity and the frequency of cloud state (blue and red lines, respectively), we
find that the daytime changes in cloud susceptibility is primarily driven by changes in the intensity of cloud
susceptibilities during the day. Additionally, as shown in Figs. 12a and b, the red lines are close to the daytime mean
values in the morning, which indicates that variations in the frequency of different cloud states have minimal impact
on changes in LWP and $\alpha_c$ susceptibilities in the morning. On the other hand, in the afternoon, both shifts in cloud
states and changes in intensities contribute to the changes in LWP and $\alpha_c$ susceptibilities. Compared with LWP and
$\alpha_c$ susceptibilities, the daytime variation of CF susceptibility shows minimal sensitivity to changes in cloud state
frequency. This limited impact stems from the fact that the daytime fluctuation in cloud state frequency is
predominantly influenced by precipitating and non-precipitating thin clouds. Meanwhile, the daytime mean CF
susceptibilities for precipitating and non-precipitating thin clouds closely align, measuring at 0.08 and 0.09,
respectively (Fig.2c). This convergence diminishes the influence of alterations in the frequency of these two cloud
states.
In summary, the daytime variation of cloud susceptibility is largely driven by the variation in its intensity.
Since polar-orbiting satellites only observe the cloud responses to $N_d$ perturbations across different cloud states at
their overpass time, they cannot fully capture the diurnal variation of cloud susceptibilities driven by the variation in
the intensity of cloud susceptibility. Given that all three cloud susceptibilities reach their daily minimum at around
13:30 LST, studies based on polar-orbiting satellite with overpass time at noon may underestimate the daily mean
value of cloud susceptibility.
**4. Discussions**
In this study, we quantify the susceptibility of warm boundary layer clouds to $N_d$ perturbation using the
pixel-level SEVIRI cloud retrievals of each time step. For heavily precipitating clouds, LWP increases under pristine
condition (e.g., $N_d < 30\ cm^{-3}$, Fig.2a). For lightly precipitating and non-precipitating clouds, LWP decreases with
$N_d$. The $N_d$-LWP relationship found in this study is consistent with that in Gryspeerdt et al. (2019) using global
mean cloud retrievals from MODIS and AMSR-E at coarser resolution of $1° \times 1°$ and daily timescale. This
consistency between different satellite measurements at different temporal and spatial scales greatly enhance our
confidence in the retrieved relationship.
This study further distinguishes non-precipitating clouds into thin and thick clouds based on their LWP. A
consistent decreasing trend in cloud water is found for both states, yet non-precipitating thick clouds exhibit more
negative LWP susceptibility ($\frac{dln(LWP)}{dln(N_d)} = -0.94$) compared to thin clouds ($\frac{dln(LWP)}{dln(N_d)} = -0.71$). The LWP
susceptibilities estimated in this study are more negative than those in Zhang et al. (2022) and Zhang and Feingold
(2023), based on similar classification of cloud states. Particularly, we found that non-precipitating thin clouds have
a decreasing trend in cloud water and a warming effect at the surface radiation while these are opposite in Zhang et
al. (2022) and Zhang and Feingold (2023). This is due to different seasons and study regions between our and their
studies. The summer boundary layer in the ENA region is deeper and less stable with higher cloud tops (e.g., Klein
and Hartmann, 1993; Ding et al., 2021; King et al., 2013) compared to the NE Pacific in Zhang et al. (2022) and the
NE Atlantic region in Zhang and Feingold (2023). The less stable condition, deeper boundary layer, and deeper
clouds could lead to a stronger cloud-top entrainment rate and result in a more negative LWP susceptibility (Possner
et al., 2020; Toll et al., 2019).
Regarding the CF adjustment to $N_d$ perturbation, a daytime mean positive response is found for
precipitating and non-precipitating thin clouds and a negative response for non-precipitating thick clouds (Fig.2c).
Few studies have quantified the CF adjustment rate at 30-minute intervals for a direct comparison of CF
susceptibility. However, similar results are found using measurements and retrievals from different platforms at
various spatial and timescales, which greatly increase our confidence in the observed CF responses toward $N_d$
perturbation. For example, using MODIS measurement, Kaufman et al. (2005) found an increase in the longitudinal
mean cloudiness for warm boundary layer clouds with increasing AOD in all four regions of the Atlantic Ocean
characterized by distinct aerosol types. Using the natural experiment of volcanic eruption at Holuhraun in Iceland,
Chen et al. (2022) found that aerosols from the eruption increase the monthly mean cloud cover by 10% over the
North Atlantic. By tracking the cloud trajectory using geostationary satellites, Christensen et al. (2020) found that
aerosol enhance both CF and cloud lifetime in the timescale of 2-3 days, especially under stable conditions. It is
worth noting that a decrease in CF was not observed in these studies, likely due to the prevalence of non-
precipitating thin clouds and precipitating clouds in the Atlantic or the NE Pacific (e.g., Zhang and Feingold, 2023)
that mask the signal from non-precipitating thick clouds without distinguishing cloud states.
Lastly, the distinct "U-shaped" daytime variation in all three cloud properties found in this study (Fig.3a-c)
is *unlikely* due to the systematic bias in $r_e$ and $\tau$ retrievals at large SZA based on the following two aspects. Firstly,
if the daytime variation is driven by retrieval bias at large SZA, we would expect the susceptibility exhibiting a
symmetric pattern at local noon. As shown in Figs. 10 and 6, the LWP and $\alpha_c$ susceptibilities for precipitating and
non-precipitating thick clouds exhibit asymmetric pattern at local noon: with a decreasing trend from 13 LST and a
daily minimum at 16 LST, and a continuously increasing trend from 11 to 18 LST, respectively. In addition, the CF
susceptibilities for all three cloud states show asymmetric patterns at local noon. Secondly, if the retrieval
uncertainty dominates the signal, we would expect less variation in cloud susceptibilities for overcast clouds, which
suffer less uncertainties in cloud retrievals from the plane-parallel assumption and the cloud 3-D effect. However,
the opposite is found from the sensitivity test where overcast clouds exhibit stronger daytime variation in cloud
susceptibilities (not shown).
**5. Conclusions**
Using $N_d$ as an intermediary variable, this study investigates the aerosol indirect effect (AIE) for warm
boundary layer clouds and its daytime variation over the ENA region with the half-hourly and 3-km cloud property
retrievals from SEVIRI on the Meteosat-11. To constrain meteorological impacts on clouds and aerosol-cloud
interaction, cloud susceptibilities are estimated within a $1° \times 1°$ grid box for each satellite time step. Based on the
daytime mean cloud susceptibilities in the LWP-$N_d$ parameter space, the sign and magnitude of cloud
susceptibilities strongly depend on the cloud states (Fig.2). Accordingly, warm boundary layer clouds are classified
into three states: precipitating clouds ($r_e$>15 μm), non-precipitating thick clouds ($r_e$<15 μm, LWP > 75 $gm^{-2}$), and
non-precipitating thin clouds ($r_e$<15 μm, LWP < 75 $gm^{-2}$).
Precipitating clouds exhibit contrasting responses in cloud LWP, with increases observed for heavily
precipitating clouds and decreases for lightly precipitating clouds. Positive $\alpha_c$ and CF susceptibilities are identified
for both heavily and lightly precipitating clouds. The net all-sky radiative forcing of the AIE on precipitating clouds
is estimated to be $-13 \; W \; m^{-2} ln(N_d)^{-1}$, with contributions from the CF and $\alpha_c$ effects of $-9.5$ and $-3.5$
$W \; m^{-2} \; ln(N_d)^{-1}$, respectively. For non-precipitating clouds, both thick and thin clouds show negative LWP
susceptibility with more negative values found for thick clouds with higher LWP and $N_d$. This is likely attributed to
the stronger shortwave absorption, larger cloud top radiative cooling rate and stronger entrainment for thick clouds.
Consistent with the evaporation-entrainment feedback, non-precipitating thick clouds exhibit a decrease in CF and
$\alpha_c$ with increasing $N_d$, and results in a net warming effect at the surface and a radiative forcing of $+4.4$
$W \; m^{-2} \; ln(N_d)^{-1}$. On the other hand, non-precipitating thin clouds show an increasing response in CF and a less
negative $\alpha_c$ susceptibility. Additionally, the radiative effect from increasing CF ($-8.3 \; W \; m^{-2} \; ln(N_d)^{-1}$) outweighs
that from a darker cloud ($+3.1 \; W \; m^{-2} ln(N_d)^{-1}$) and leads to a net cooling effect of $-5.2 \; W \; m^{-2} \; ln(N_d)^{-1}$.
Warm boundary layer clouds manifest distinct and significant ($p<0.05$) daytime variations in LWP, $\alpha_c$, and
CF susceptibilities. All three cloud susceptibilities exhibit "U-shaped" diurnal patterns with clouds being less
susceptible in the morning and evening and more susceptible at local noon (Fig.3).
Daytime variation in LWP and $\alpha_c$ susceptibilities is likely attributed to the transition in cloud state while
clouds sustaining the memory of responses to $N_d$ of the previous state (H1 in Table 1). From morning to noon, with
increasing solar radiation, non-precipitating thick clouds evolve to thin clouds. Thin clouds decayed from thick show
significantly more negative LWP and $\alpha_c$ susceptibilities than thin clouds that are previously thin (Fig.5). Therefore,
LWP and $\alpha_c$ susceptibilities decrease from morning to noon for thin clouds and reach their daily minima at noon
(Fig.4). In the afternoon, thin clouds develop to thick clouds while retaining the memory of less susceptible to $N_d$
perturbations (Fig.7) and therefore leads to an increase in LWP and $\alpha_c$ susceptibilities for nonprecipitating thick
clouds in the afternoon (Fig.6). Meanwhile, daytime variation in CF susceptibility for non-precipitating thick clouds
is more likely driven by changes in cloud morphology rather than the transition of cloud state (Fig.8, H3 in Table 1).
Compared to non-precipitating clouds, precipitating clouds exhibit smaller daytime variation in cloud susceptibility
(Fig.10).
The daytime variation in cloud susceptibility is primarily driven by changes in the intensity of cloud
susceptibilities from morning to noon, rather than changes in the frequency of occurrence of different cloud states
(Fig.12). As the polar-orbiting satellites only observe cloud susceptibilities across different cloud states during a
specific overpass time, they overlook the change in the intensity of cloud susceptibilities during the day. More
specifically, based on the daytime variation of cloud susceptibilities found in this study, LWP susceptibility
estimated at 13:30 LST could underestimate the daytime mean value by 26.3% ($-0.76$ compared to $-0.60$),
underestimate the $\alpha_c$ susceptibility by 475% ($-0.023$ compared to $-0.004$), and underestimate the CF susceptibility
by 120% ($-0.019$ compared to $+0.055$). It is worth noting that both the daytime variation and the daytime mean
values of cloud susceptibilities in this study are estimated based on the regression analysis on spatial data within
each satellite time step, with the assumption that the temporal change of cloud properties from $N_d$ perturbations can
be represented by the spatial relationships.
This study underscores the importance of considering the diurnal cycle of cloud susceptibilities when
quantifying AIE and their impacts on clouds and radiation. The classification of cloud states enables us to
distinguish the sign, magnitude, and underlying processes driving the diurnal variation of AIE.
To further advance our understanding of the diurnal variation of AIE, several avenues for future research
can be pursued. Firstly, it is important to address uncertainties associated with satellite retrievals, which can
propagate into uncertainties in the retrieved $N_d$, as discussed in Grosvenor et al. (2018). Future study could utilize
active sensors to reduce these uncertainties, particularly during nighttime conditions. Moreover, using the retrieved
$N_d$ as a proxy of aerosol concentration may introduce uncertainties related to cloud processes that can act as sources
or sinks of $N_d$, potentially buffer the relationships between $N_d$ and cloud condensation nuclei. Future investigations
are needed to better understand the relationships, and how they vary with different cloud processes and throughout
the day. Lastly, this study encompasses all warm boundary layer clouds without considering the highly diverse
meteorological regimes and the associated cloud types in the ENA region. Classification of the synoptic and
meteorological conditions associated with different cloud states and aerosol properties would contribute to a more
comprehensive understanding, allowing for the disentanglement of the impacts of meteorology from AIE.


**Data availability:**
SEVIRI Meteosat-11 cloud retrieval products, produced by NASA LaRC SatCORPS group, are available from the
Atmospheric Radiation Measurement (ARM) Data Discovery website at https://adc.arm.gov/discovery/, Minnis
Cloud Products Using Visst Algorithm (visstgridm11minnis). The ARM ground-based radar and lidar observations
are available from ARM Data Discovery, KAZRARSCL, (arsclkazr1kollias).

**Acknowledgment:**
We are grateful to the Atmospheric Radiation Measurement (ARM) user facility, a U.S. Department of Energy
(DOE) Office of Science user facility managed by the Biological and Environmental Research Program for
providing ARM observation data and archiving SEVIRI Meteosat-11 cloud retrieval products. We mainly used the
computing resources from the National Energy Research Scientific Computing Center (NERSC), which is supported
by the Office of Science of the U.S. Department of Energy under Contract No. DE-AC02-05CH11231. We
appreciate the constructive comments from two anonymous reviewers that helped improve the manuscript. This
work was performed under the auspices of the U.S. DOE by LLNL under contract DE-AC52-07NA27344. (LLNL-
JRNL-851496)

**Financial support:**
This work is supported by the DOE Office of Science Early Career Research Program and the ASR Program. DP
acknowledges the support of the NASA CloudSat CALIPSO Science Recompete Program.

**Competing interests:**

The authors declare that they have no conflict of interest.

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
