# Peer review of "Daytime variation of aerosol indirect effect for warm marine boundary layer clouds in the eastern north Atlantic"

_EGUsphere, 2023_

## Referee Comment (RC1)

The authors present a detailed study of cloud property susceptibilities (of liquid water, albedo, and cloud fraction) to droplet numbers in low liquid marine clouds in the NE Atlantic, using new geostationary satellite data. They show diurnal variations in the susceptibilities, with the responses being most negative in the early afternoon – when the widely used Aqua satellite collects data. This could point at a bias in studies quantifying the susceptibilities from this time of day. The study is carefully argued, with the right level of detail, and gives important insights into the responses of clouds and climate to anthropogenic aerosol. But there are a number of major points to be addressed before publication.

Major comments:
1. Arola et al (2022) showed that even a positive Nd-LWP relationship can give negative correlations when regressing, due to natural variability and retrieval errors. Is this a pertinent potential source of error for your study? They restrict to small retrieval errors (working on MODIS data) and show that the relationships grow weaker. Please test the sensitivity of your results, especially Figure 3 on diurnal variability, to restricting more or less to retrieval uncertainty. Is there a way you can assess the importance of natural variability changing the retrieved slope?

2. On causality: The retrievals will show causation and covariation: position in the cloud (edge vs. inner part, time variations)? You argue, that we do not expect the meteorological parameters to vary much in the 1deg grid box, but then we expect the same for aerosol concentrations–can we still say that Nd variations are from aerosols alone? This is what is implied in calling the correlations/regression coefficients 'susceptibilities'. To what extent are we able to say that an (e.g. anthropogenic) increase in Nd would lead to a corresponding change in LWP, albedo, CF? Please add a discussion of this.

3. This becomes particularly important when looking at transitions or 'cloud memory': Does an observed stronger correlation of Nd and LWP in thin clouds which were previously thick mean, that if we now added aerosol, they would dry even more? Or is this more negative correlation because of the processes involved in the thinning of thick clouds? On page 13, regarding your hypothesis 3, you examine the decay of thin clouds, which does not explain the changes. But clouds undergoing a similar 'decay' from thick to thin are shown to have strong susceptibilities. In particular, in line 405, you say that "Similar results are obtained using classification methods based on [different CF thresholds (e.g., from 10% to 30%) or] changes in the mean LWP". To test hypothesis 2 (cloud memory), you use a two-hour LWP change classification. This means that the only difference for testing hypotheses 3&2 is the time scale (30 mins vs. 2h), but it is not immediately obvious why dissipation should not happen more slowly. In short: Could the cloud memory effect (thick-thin) just be dissipation of thick clouds, and the mechanism not cloud memory but covarying effective radii and LWP? To claim 'cloud memory', you need to rule this out.

Specific comments:

l. 308-310: *"Precipitation acts to stabilize the boundary layer, remove water from cloud top, and reduce the entrainment rate (Sandu et al., 2007, 2008). Precipitation suppression and entrainment weakening work in concert and result in a net increase in LWP with increasing $N_d$."*

This does not make sense to me. If precipitation tends to reduce the entrainment rate, its suppression strengthens the entrainment rate. So both processes work against each other? Please, could you elaborate on this?

l. 334-336: *"The opposite signs in LWP and CF susceptibilities for non-precipitating thin clouds cannot be solely explained by the evaporation-entrainment feedback. In the next section, two additional hypotheses regarding the development/dissipation of clouds and the transition of cloud states will be tested."*

I agree that the opposite signs in LWP and CF adjustment cannot be explained with evaporation-entrainment. But I do not find an explanation later, after you introduce the hypotheses and conclude susceptibility variations are due to cloud memory. To me, this does not explain the opposing signs. Please include a discussion.

L 368: *"The diurnal variation of cloud susceptibility is statistically significant at a 95% confidence level based on a student's t-test."* Please elaborate: What are the variables/means that are compared here, with what variability?

Figure 2: What is the variability (e.g. the standard error of the mean) in the averaged 1x1deg susceptibilities for each LWP-Nd bin? Can you state which average susceptibilities are significant? Why are some squares missing in some panels but not in others?

l. 616: *"The less stable condition over the studied region leads to a deeper boundary layer, deeper clouds, and a stronger entrainment rate at the cloud top, all of which may cause a more negative LWP susceptibility"*

To me, this raises the question how much can we extrapolate to all year liquid marine clouds from July data. If the susceptibility is a function of boundary layer depth (which it should be), then it will change with changing SSTs over the seasonal cycle. Please discuss.

Technical corrections:
L 40/41: The cloud adjustment effect, on the other hand, is highly variable
L 119: We then discuss (delete comma)
L 281 responses from both for
L 370 "valuers"
L 514 increases in the afternoon, and becomes (instead of become)

Reference
Arola, A., Lipponen, A., Kolmonen, P., Virtanen, T. H., Bellouin, N., Grosvenor, D. P., Gryspeerdt, E., Quaas, J., & Kokkola, H. (2022). Aerosol effects on clouds are concealed by natural cloud heterogeneity and satellite retrieval errors. *Nature Communications*, *13*(1), 7357. https://doi.org/10.1038/s41467-022-34948-5

---

## Referee Comment (RC2)

This study provides a comprehensive look at daytime evolution of low cloud susceptibility to droplet number ($N_d$) perturbations. These susceptibilities are derived from snapshots of cloud and radiation fields taken from passive sensors (SEVIRI), onboard a geostationary satellite (Meteosat11), within a 10-by-10-degree box over the NE Atlantic region. The authors found persistent "U-shape" evolutions in cloud susceptibility, including cloud water, cloud albedo, and cloud fraction, from which they argue that polar-orbiting satellite derived cloud susceptibility underestimate the daytime-mean cloud susceptibility. They attribute the observed evolution in cloud susceptibility to two main hypotheses that they formulated and tested: i) clouds have "memory" of their past susceptibility signal associated with the cloud state they transition from; ii) cloud responses to $N_d$ perturbations have a longer timescale than the satellite observing time interval (30 mins).

Overall, I found the study interesting and intriguing in some sense. Clearly, the authors have put in lots of thoughts on this problem and lots of efforts in interpreting these results. The manuscript is nicely constructed and organized. That being said, I do think there are some key/fundamental issues that need to be addressed and justified first. If these concerns and issues can be sufficiently addressed, I believe this work has the potential to provide new, impactful insights and make significant contribution to the field.

**Major concerns/comments:**

1. The authors adopted a methodology for deriving cloud susceptibility that uses spatial regression of cloud properties to $N_d$ within satellite snapshots. This is done to minimize the impact from confounding meteorology. I found the extension of this approach to cloud fraction susceptibility is a bit unjustified, and I'm in general concerned about the appropriateness of applying this methodology to cloud fraction susceptibility. Cloud fraction is a tricky quantity by definition. It depends strongly on the spatiotemporal scale of one's investigation, for example, the distribution of daily cloud fraction will look very different from that of monthly cloud fraction at any given location on Earth. The degree of spatial aggregation also strongly affects cloud fraction, e.g., cloud fraction reduces to binary (zero and one) at pixel level. Overall, I find the discussion/explanations around CF susceptibility quite vague and often led to confusion, which could potentially be related to an ill-defined methodology. Some of my specific concerns are:
    a. How confident are you that, at 25-km resolution, variations in CF is not simply coming from the geometry of the cloud field, such that one scene captures the majority of a cloud while another scene captures only a small portion of the same cloud, resulting in a large difference in cloud fraction between the two scenes, while $N_d$ has nothing to do with it.
    b. On lines 335-336, the readers are directed to the next section for explanations on the CF susceptibility pattern seen in Fig. 2c. In the next section, your discussion and hypotheses testing are targeted at CF susceptibility evolutions, while I'm still missing an explanation for the non-precip thin cloud that exhibit potentials in cloud fraction enhancement.
    c. Fig. 11 is another example that I am having difficult times wrapping my head around. While Fig. 11a and b suggest that the frequency of occurrence of cloud state in the LWP-$N_d$ space is indeed changing throughout the day, the null contribution from

frequency changes shown in Fig. 11c troubles me a bit. I guess it's related to the pattern we see in Fig. 2c, which I still struggle to decipher.

2. Regarding the explanation for the "U-shape" evolutions, it seems a bit suspicious to me that all 3 quantities show the same shape of evolution. Moreover, you use cloud transition categories to explain the observed U-shape evolution, yet you see the similar "U-shape" for individual categories, suggesting to me that there are something fundamental that has not been teased out. This makes wonder if this is due to retrieval artifacts associated with relatively high solar zenith angle during early morning and late afternoon? Smally and Lebsock (2023) made a comparison between retrieved LWP from tau & $r_e$ using adiabatic assumption with microwave imager retrieved LWP, onboard geostationary satellites. They found non-negligible bias even at intermediate SZA. The same bias applies for $N_d$ retrievals as well, in a more pronounced way I suppose, given the high order dependence of $N_d$ calculation on $r_e$. I wonder if this "U-shape" evolution can be explained by tau & $r_e$ retrieval biases as a function of SZA. To me, even small biases can produce the trend you see towards higher SZAs (i.e. towards early morning and late afternoon). I suppose one can play with some synthetic data to rule this possibility out.

3. Assuming daytime evolution in cloud susceptibility is not SZA-dependent for now, an alternative explanation for the "U-shape" evolution is the change in meteorology, i.e., daytime evolution in boundary layer (BL) depth. From morning to early afternoon, BL deepens and becomes decoupled, enhancing the role of entrainment feedbacks in governing cloud susceptibility. In the afternoon, BL re-couples. This could possibly give you the "U-shape" evolution in cloud susceptibility. I wonder if the authors have considered this as an alternative explanation and tested it?

4. Regarding the "memory" hypothesis, I have to say that I'm quite lost in this argument, by the design of the analysis, by the inconsistency that I spotted when reasoning with this argument, and by the actual physical meaning of "memory" in the context of cloud susceptibility.
   a. Fundamentally, the concept of "memory" requires a Lagrangian perspective, however, I believe all the analyses done in this study are based on a Eulerian framework. This means that the "memory" in this study is actually the domain "memory" of the clouds that were previously present in the domain, not cloud "memory" of its past states. I believe one needs to at least do a grid-box tracking, if not individual cloud tracking, to support this argument, or show that clouds are stationary/semi-stationary in the study region.
   b. One example of the inconsistency that I observed is that in Fig. 5, if I follow the argument correctly, thick→thin transition will lead to more negative LWP susceptibility based on the LWP-$N_d$ plot, however, at early morning where thick→thin is the most frequently occurring type, the overall LWP susceptibility is the highest (Fig. 4b). Furthermore, how do you explain the "U-shape" in the evolution that is specific to thick→thin transition clouds, they all have the same type of "memory" according to your argument.
   c. I wonder at what timescale do you expect these clouds to detach/decorrelate from their "memory" of past states? Existing literature seems to suggest a decorrelation timescale that is much shorter than your 2-hr window, on the order of 10 to 15 mins, for marine

boundary layer clouds. I wonder if you could elaborate on how does this decorrelation timescale fit in your 2-hr "memory" hypothesis?

    d.  I'm familiar with the concept that cloud have "memory" of their near-past state, in which clouds carry-over their past tendencies for a short period of time. However, I am afraid that I don't quite follow the argument that clouds have a "memory" of its past 'susceptibility.' I wonder if you could elaborate more on the physical mechanism of it. I believe the schematic (Fig. 8) you shown refers to the concept I mentioned, such that stronger evaporation & entrainment are carried over if the cloud just transitioned from thick to thin. It speaks to the temporal evolution of clouds, but how does this work for a "memory" of susceptibility, which is essentially a spatial regression, is less clear to me.

**Minor comments:**

- Line 36-37, 600, 669-671, the authors made the point that susceptibility derived from polar-orbiting satellites, Aqua in particular, underestimates the daily mean value of cloud susceptibly. While this is true based on the findings from this study, I would like to raise the point that this underestimation does not necessarily translate into an underestimation of the actual cloud response, as the simple arithmetic mean of susceptibility ('local derivatives') is not the same as a time integral of cloud responses, which is more relevant to the scaling up of radiative effects of ACI.
- Line 48, I guess you need more reference here than just Albrecht 1989, as it only speaks to the precip-suppression effect.
- Line 109, note that Gryspeerdt et al. (2021) did not use geostationary satellite but rather polar-orbiting satellites, i.e., Terra and Aqua.
- Line 114, I thought you did not "track" any cloud or cloud field and used with the Eulerian framework, I would be more precise here to avoid confusion.
- Line 137, reference for LWP calculation formular?
- Line 155, $c_w$ is a function of pressure as well.
- Line 172-173, did you average tau and $r_e$ first and then calculate $N_d$ or calculate $N_d$ first then average? This difference can induce huge bias in albedo susceptibility quantification as shown in Feingold et al. (2022, ACP).
- Line 175, note Zhou et al. (2021) used 2x2 grid box, if I recall correctly, worth double checking.
- Line 187, why cloudy pixels at cloud edge are set as clear?
- Line 218-219, why you opt for a Eulerian framework here? Isn't a Lagrangian framework the appropriate choice here as you want to investigate cloud "memory"?
- Line 254, what do you mean by "witnessed by clouds"?
- Line 266, I wouldn't use "hypothesis" here as these entrainment feedback mechanisms have been previously established, using large-eddy simulations.
- Line 280-281, what do you mean by "meteorological influences on clouds likely dampen the signal of the AIE…" I feel like you try to say meteorological confounding tends to obscure the AIE signal, probably need to rephrase here.
- Line 352-353, I don't think you have diurnal cycles, only daytime, right?
- Fig. 3c, the day-to-day (or spatial variation within the 10 by 10 box) variation is huge, at a given time of the day, is it possible to tell the sign of CF susceptibility with statistical significance?

- Line 439-440, I don't follow this argument, your susceptibility for each individual transition group shouldn't be affect by their relative frequency of occurrence at a given time, if I understand your method correctly, then why fewer samples in the thick/precip→thin groups lead to less difference between the groups?
- Line 454-456, again, why not using Lagrangian tracking in this assessment?
- Table. 1, I wonder if there are more effective ways of illustrating hypotheses than showing them in a table, the current structure of the table is quite difficult to decipher, at least for me. In particular the N/A boxes confuse me, and I don't see discussion of them in the main text.
- Line 478-481, these arguments are vague, if your "memory" argument stands, shouldn't we see a lagged evolution in thin clouds compared to thick clouds, instead of similar evolution?
- Line 488, why this is particularly prominent in the morning, I could not think of a reason following your "memory" argument.
- Line 567, worth repeating the question here.
- Fig. 10b, why the difference between thin→rain and thick→rain is reversed during midday, compared to early morning?
- Line 602, "instantaneous responses of warm boundary layer clouds…" I suggest rephrasing, as what you shown in the study is basically correlations (regression slopes), not actual responses (which refer to perturbation experiments).
- Line 621, "instantaneous CF adjustment rate" is a bit awkward, I suggest rephrasing (see comment above).
- Line 643, again, established mechanisms, not hypotheses.
- Font size too small in Figs. 3-7, 9-10. I suggest matching them to that of Fig. 11, which looks just fine.
- In Fig. 3-7, 9-11, basically all time-series figures, I strongly recommend replacing the horizontal lines indicating daytime mean values with lines indicating "0", and one can simply label the daytime mean values on each panel. The current horizontal lines are so seductive to be seen as the zero lines.

**Reference**

- Smalley, K. M., and M. D. Lebsock, 2023: Corrections for Geostationary Cloud Liquid Water Path Using Microwave Imagery. J. Atmos. Oceanic Technol., 40, 1049–1061, https://doi.org/10.1175/JTECH-D-23-0030.1.
- Feingold, G., Goren, T., and Yamaguchi, T., 2022: Quantifying albedo susceptibility biases in shallow clouds, Atmos. Chem. Phys., 22, 3303–3319, https://doi.org/10.5194/acp-22-3303-2022.

---

## Author Comment (AC1)

**Responses to reviewer comments**
The original comments are in *blue italic font*, and our response are in black font.

**Reviewer #1:**
*The authors present a detailed study of cloud property susceptibilities (of liquid water, albedo, and cloud fraction) to droplet numbers in low liquid marine clouds in the NE Atlantic, using new geostationary satellite data. They show diurnal variations in the susceptibilities, with the responses being most negative in the early afternoon – when the widely used Aqua satellite collects data. This could point at a bias in studies quantifying the susceptibilities from this time of day. The study is carefully argued, with the right level of detail, and gives important insights into the responses of clouds and climate to anthropogenic aerosol. But there are a number of major points to be addressed before publication.*

We appreciate the positive assessment and the constructive suggestions. Below, we provide a point-by-point response to each comment and details of the modifications made to the text and figures to address these points.

*Major comments:*
*1 Arola et al (2022) showed that even a positive Nd-LWP relationship can give negative correlations when regressing, due to natural variability and retrieval errors. Is this a pertinent potential source of error for your study? They restrict to small retrieval errors (working on MODIS data) and show that the relationships grow weaker. Please test the sensitivity of your results, especially Figure 3 on diurnal variability, to restricting more or less to retrieval uncertainty. Is there a way you can assess the importance of natural variability changing the retrieved slope?*

Thank you for your insightful question and for providing the reference paper. As shown in Arola et al. (2022), Feingold et al. (2022), and Zhou and Feingold (2023), small-scale cloud heterogeneity, spatial scale of the cloud organization, or even the $N_d$ retrieval algorithm led to large variabilities in the retrieved LWP and albedo susceptibilities. To account for these factors, aggregation/smoothing of the pixel-level data before the regression is performed (e.g., Feingold et al., 2022; Zhou and Feingold, 2023). In our study, we first smoothed the pixel-level Meteosat retrievals to a 0.25° spatial resolution and aggregate all the data over a $10° \times 10°$ domain and 4 years. We have included references to these papers in section 2 to provide a clear explanation of our approach and to emphasize our efforts to mitigate potential biases arising from natural variability. Moreover, before performing the smoothing and aggregation, we removed pixels near cloud edges, which should substantially reduce biases in cloud retrievals.

To assess the sensitivity of our results to Meteosat retrieval uncertainty, given the absence of true values for the retrieved $\tau$ and $r_e$, we manually applied scaling factors of 120% and 80% to the retrieved $\tau$ and $r_e$. As shown in Figures R1 and R2, consistent daytime variation of cloud susceptibility is retained for all four variables. However, the magnitude of change for each variable is sensitive to the retrieved $\tau$ and $r_e$ as expected. We have added discussion regarding to the influence of retrieval uncertainty and cloud heterogeneity to the quantified susceptibility in section 2 as follow:

"As found by Arola et al. (2022) and Zhou and Feingold (2023), the retrieved cloud susceptibilities are sensitive to small-scale cloud heterogeneity, the co-variability between cloud

properties and $N_d$, and the spatial scale of cloud organization. To reduce the biases resulting from heterogeneity and co-variability, we first average the 3-km pixel-level cloud retrievals to a regular $0.25° × 0.25°$ grid for each half-hourly time step. As suggested by Feingold et al (2022), $N_d$ retrieval was performed at pixel-level using Eq. (1), and then averaged to a $0.25°$ resolution." (Lines194-199).

[Figure]

Figure R1. Same as Figure 3, but with retrieved $\tau$ and $r_e$ scaled up by 120%.

[Figure]

Figure R2. Same as Figure 3, but with retrieved $\tau$ and $r_e$ scaled down by 80%.

*2. On causality: The retrievals will show causation and covariation: position in the cloud (edge vs. inner part, time variations)? You argue, that we do not expect the meteorological parameters to vary much in the 1deg grid box, but then we expect the same for aerosol concentrations–can we still say that Nd variations are from aerosols alone? This is what is implied in calling the correlations/regression coefficients 'susceptibilities'. To what extent are we able to say that an (e.g. anthropogenic) increase in Nd would lead to a corresponding change in LWP, albedo, CF? Please add a discussion of this.*

The scale of heterogeneity and variability in aerosols is typically much smaller than that of meteorological and synoptic conditions. Within a $1° \times 1°$ box, variations in aerosol and $N_d$ can arise from sources like occasional ship emissions, local island emissions, and long-distance transportation. It's acknowledged that, according to the definition of aerosol-cloud interactions in this study, it inevitably comprises the response of cloud properties to $N_d$ perturbation (the targeted signal) and the spatial/temporal covariation between $N_d$ and cloud properties.

To minimize the influence from temporal covariation, we quantify the susceptibility within each time step of satellite observations, similar to Figure 1b verses 1a in Arola et al. (2022). As discussed in the previous comment, to minimize the influence of spatial covariation, we averaged the pixel-level satellite retrievals to a 0.25° spatial resolution and aggregate the data over a $10° \times 10°$ domain for a period of 4 years. However, unlike well-controlled numerical

simulations, this observational study alone cannot quantify the contribution from aerosol-cloud interaction versus from covariation or establish causality in the observed relationship. Future studies are discussed in the manuscript to address this challenge.

3. *This becomes particularly important when looking at transitions or 'cloud memory': Does an observed stronger correlation of Nd and LWP in thin clouds which were previously thick mean, that if we now added aerosol, they would dry even more? Or is this more negative correlation because of the processes involved in the thinning of thick clouds? On page 13, regarding your hypothesis 3, you examine the decay of thin clouds, which does not explain the changes. But clouds undergoing a similar 'decay' from thick to thin are shown to have strong susceptibilities. In particular, in line 405, you say that "Similar results are obtained using classification methods based on [different CF thresholds (e.g., from 10% to 30%) or] changes in the mean LWP". To test hypothesis 2 (cloud memory), you use a two-hour LWP change classification. This means that the only difference for testing hypotheses 3&2 is the time scale (30 mins vs. 2h), but it is not immediately obvious why dissipation should not happen more slowly. In short: Could the cloud memory effect (thick-thin) just be dissipation of thick clouds, and the mechanism not cloud memory but covarying effective radii and LWP? To claim 'cloud memory', you need to rule this out.*

Thank you for your insightful question. We have explored different definitions for the dissipation and development of clouds. As shown in Figure S4, when using the change of cloud fraction (CF) as the definition of cloud dissipation and development, non-precipitating thin clouds with increasing or decreasing CF have less negative LWP susceptibilities than clouds with constant CF. Therefore, the decrease of LWP susceptibility from morning to noon is unlikely due to the dissipation and development of thin clouds.

We also tried defining the dissipation and development of clouds using the change of cloud LWP, as shown in Figure R3. Dissipating/developing clouds are defined as clouds with a decrease/increase in LWP greater than 25 $gm^{-2}$ in the past two hours. Results are consistent with different LWP thresholds from 15-45 $gm^{-2}$ (not shown). To compare with the definition of the cloud state transition, a same two-hour time window was applied, which is different from the 30-min time window for Figure S4. As the non-precipitating thick and thin clouds are defined by cloud LWP, results shown in Figure R3 are similar to results shown in Figure 5. Dissipating clouds with decreasing LWP have more negative LWP susceptibility than clouds with constant LWP (Figure R3b). However, differences between dissipating and constant clouds are not statistically significant. This is likely because the decrease of cloud LWP does not distinguish between the transition from non-precipitating thick to thin clouds or the transition from precipitating to non-precipitating thin clouds.

In summary, non-precipitating thin clouds with decreasing LWP retain the "memory" of cloud susceptibility of their previous conditions (Figure 5 and Figure R3). Meanwhile, different transitions in cloud state (e.g., rain → thin vs. thick → thin) show significantly different cloud susceptibilities. Furthermore, the dissipation/development of clouds include change in both cloud LWP and CF, while changes in CF cannot explain the evolution of cloud LWP susceptibility (Figure S4). Therefore, daytime variation of cloud LWP susceptibility is better explained by Hypothesis 1 instead of by Hypothesis 2 in Table 1. We have added a discussion on the different definitions of cloud dissipation and a discussion of Fig. R3 to the manuscript.

"Besides the change in CF, dissipation/development of clouds can be defined by change in LWP. However, as our definition of thin and thick clouds use LWP thresholds, results based on change in LWP are similar to results shown in Fig. 5, but with weaker signal (not shown). This indicates that classification of precipitating verses non-precipitating clouds is necessary in distinguishing cloud responses to $N_d$ perturbations than merely using the LWP threshold." (Lines 501-505).

[Figure]

Figure R3. Daytime variation of non-precipitating thin clouds that have small changes in the 1° × 1° mean LWP (No change, solid line with circle symbols), with an increase in LWP (developing, solid line with triangle symbols), and with a decrease in LWP (dissipating, dash line with diamond symbols) within a two-hour window. (a) Percentage of occurrence for the three groups above, (b) cloud LWP susceptibility ($dln(LWP)/dln(N_d)$), (c) cloud albedo susceptibility ($d\alpha_c/dln(N_d)$), and (d) cloud fraction susceptibility ($dCF/dln(N_d)$) for non-precipitating thin clouds. Symbols representing different cloud stages are noted in (b). In (b)-(d), filled markers indicate data points that are significantly different from the other two groups (p<0.05). Open markers indicate statistical insignificance.

*Specific comments:*
*-L308-310: "Precipitation acts to stabilize the boundary layer, remove water from cloud top, and reduce the entrainment rate (Sandu et al., 2007, 2008). Precipitation suppression and*

*entrainment weakening work in concert and result in a net increase in LWP with increasing Nd."*

*This does not make sense to me. If precipitation tends to reduce the entrainment rate, its suppression strengthens the entrainment rate. So both processes work against each other? Please, could you elaborate on this?*

Thank you for pointing out this discrepancy. Because of the influence of precipitation on boundary layer and cloud water, in heavily precipitating clouds, the entrainment rate is smaller compared to that in non-precipitating clouds with similar LWP. With increasing $N_d$ from aerosol perturbations, cloud droplet sizes decrease and suppress precipitation, causing an increase in cloud LWP. Meanwhile, the increased aerosols also enhance cloud top entrainment rate, leading to a decrease in cloud LWP. As the entrainment is weaker in heavily precipitating clouds, the precipitation suppression feedback likely outweighs the entrainment feedback, resulting in a net increase in LWP (e.g., Chen et al., 2014; Toll et al., 2019)). Modifications were made in the tracked changes for clarity: "Therefore, precipitating clouds exhibit smaller entrainment rate than non-precipitating clouds with similar LWP. The increase of LWP from precipitating suppression feedback outweighs the decrease of LWP from entrainment feedback and results in a net increase in LWP (e.g., Chen et al., 2014; Toll et al., 2019)." (Lines: 363-365).

*-L334-336: "The opposite signs in LWP and CF susceptibilities for non-precipitating thin clouds cannot be solely explained by the evaporation-entrainment feedback. In the next section, two additional hypotheses regarding the development/dissipation of clouds and the transition of cloud states will be tested."*

*I agree that the opposite signs in LWP and CF adjustment cannot be explained with evaporation-entrainment. But I do not find an explanation later, after you introduce the hypotheses and conclude susceptibility variations are due to cloud memory. To me, this does not explain the opposing signs. Please include a discussion.*

Increases in aerosol decrease cloud drop size and increase cloud drop number concentration near cloud top, which enhance the cloud top radiative cooling rate and the entrainment rate. The enhanced radiative cooling and entrainment induce downdraft and mixing from cloud top (e.g., Xue and Feingold, 2006). These factors contribute to destabilize the boundary layer and facilitate moisture transport from the ocean surface to clouds, thus enhance new cloud formation and extend cloud lifetime (e.g., Christensen et al. 2020). This hypothesis is consistent with and supported by the relative low CF for these clouds (Fig. S2a) and the diurnal variation in LWP susceptibility for non-precipitating thin clouds (Figure 4c). In the morning, the boundary layer is typically shallower and well-mixed with clouds coupled to the surface. Therefore, the CF susceptibility for thin clouds is large positive in the morning, and gradually decreases from morning to noon. However, the near zero CF susceptibility in the afternoon is not supported by this hypothesis. Further analyses and model simulations are needed to better understand the diurnal evolution of aerosols' impact on entrainment rate, boundary layer state, cloud cover and lifetime to explain the observed daytime variation of CF susceptibility for non-precipitating thin clouds. The opposite signs observed in LWP and CF susceptibility suggest that the aerosol indirect effect likely redistributes cloud water horizontally, causing clouds to become thinner and wider. A related discussion has been added to Figure 2 on the daytime mean CF susceptibility and to Figure 5 on the daytime variation of CF susceptibility as follow.

"A possible explanation for the increased CF is the enhanced cloud top radiative cooling rate help to mix the boundary layer facilitate moisture transport from the ocean surface to cloud, and therefore favor new cloud formation and extend cloud lifetime (e.g., Christensen et al. 2020). This hypothesis is consistent with and supported by the relative low CF for these clouds (Fig. S2a) and the diurnal variation in LWP susceptibility for non-precipitating thin clouds, which will be discussed in the next section. The opposite signs of LWP and CF susceptibilities indicate that the AIE might redistribute cloud water horizontally and make the thin clouds thinner and wider." (Lines 405-410).

"As the CF susceptibility for thin clouds transitioned from precipitating clouds and thick clouds greatly decrease from morning to noon, the CF susceptibility for thin clouds decrease from large positive to near zero from morning to noon (Fig. 4c). Another possible explanation on the evolution of CF susceptibility is the influence of aerosols on boundary layer mixing and the evolution of boundary layer from morning to noon. The enhanced entrainment rate and radiative cooling rate from $N_d$ perturbations help to destabilize the boundary layer and transport moisture from the ocean surface to clouds, which facilitate new cloud formation. As the boundary layer is typically well mixed in the morning with clouds coupled to the surface, this impact is strongest in the morning and gradually decrease from morning to noon. In the afternoon, on the other hand, thin clouds transition from all three states have near-zero CF responses to $N_d$ perturbations, which cannot be explained by the hypothesis above. Further analyses and model simulations are needed to better understand the diurnal evolution of aerosols' impact on entrainment rate, boundary layer state, cloud cover and lifetime to explain the observed daytime variation of CF susceptibility for non-precipitating thin clouds." (Lines 549-560)

*-L368: "The diurnal variation of cloud susceptibility is statistically significant at a 95% confidence level based on a student's t-test." Please elaborate: What are the variables/means that are compared here, with what variability?*

We compared all four cloud susceptibilities at different times of the day. The results indicate that cloud susceptibilities in the morning and evening exhibit a statistically significant difference compared to those at noon, established at a 95% confidence level. We also did a trend analysis of the daytime mean value of the cloud susceptibility, which is statistically significant. The high variability in cloud susceptibility includes both spatial and temporal variability, which highlights the complex interplay between synoptic conditions that varies diurnally and cloud states in the ENA region.

*-Figure 2: What is the variability (e.g. the standard error of the mean) in the averaged 1x1deg susceptibilities for each LWP-Nd bin? Can you state which average susceptibilities are significant? Why are some squares missing in some panels but not in others?*

The variability of the cloud susceptibilities for each $N_d$ and LWP bin is shown in Figure R4. The difference in cloud susceptibility for different cloud states are statistically significant for all four variables. The blank bins in Figure 2 are bins with sample number smaller than 100. Discussion on the variability of the 1° cloud susceptibilities have been added to section 3.2 in the manuscript.

"The variability of LWP susceptibility in different LWP-$N_d$ bins vary between 0.4 to 1.2 (not shown), while the LWP susceptibilities for precipitating clouds are statistically significant than other two cloud states at a 95% confidence level." (Lines 354-355)

"The variability in the overall 1° $\alpha_c$ and CF susceptibilities range between 0.05-0.15 and 0.3-0.6, respectively (not shown). The $\alpha_c$ and CF susceptibilities for precipitating clouds are statistically significant than other two cloud states at a 95% confidence level." (Lines 375-376)

"Non -precipitating thick clouds exhibit the greatest variabilities in their LWP and $\alpha_c$ susceptibilities in the three cloud states. The 1° LWP and $\alpha_c$ susceptibilities vary between 1.0-2.4 and 0.2-0.25, respectively (not shown). Due to the enhanced entrainment and evaporation, the mean CF mostly decreases with increasing $N_d$, with mean CF susceptibilities ranging from −0.1 to +0.04 $ln(N_d)^{-1}$ (Fig. 2c). Variability in CF susceptibilities for non-precipitating thick clouds is larger than that for precipitating clouds and smaller than the non-precipitating thin clouds (not shown)." (Lines 389-393)

[Figure]

Figure R4. Difference between the upper 75th percentile and the lower 25th percentile of cloud susceptibilities for different $N_d$ and LWP bins during the daytime. (a) cloud LWP susceptibility ($dln(LWP)/dln(N_d)$), (b) cloud albedo susceptibility ($d\alpha_c/dln(N_d)$), (c) cloud fraction susceptibility ($dCF/dln(N_d)$), (d) cloud shortwave susceptibility ($-dSW_{TOA}^{up}/dln(N_d)$) weighted by the frequency of occurrence of samples of each bin.

*- L616: "The less stable condition over the studied region leads to a deeper boundary layer, deeper clouds, and a stronger entrainment rate at the cloud top, all of which may cause a more negative LWP susceptibility"*

*To me, this raises the question how much can we extrapolate to all year liquid marine clouds from July data. If the susceptibility is a function of boundary layer depth (which it should be), then it will change with changing SSTs over the seasonal cycle. Please discuss.*

The LWP and albedo susceptibilities should be less negative and more positive, respectively, in other seasons compared to summer in the ENA region. The boundary layer is shallower and less stable in winter, spring, autumn than in summer, and there are more cumuli, convective and heavily precipitating clouds and less stratiform clouds in the boundary layer. All of these factors lead to larger LWP and albedo susceptibilities. The annual mean values should be close to Zhang and Feingold (2023) study. The following discussion has been added to the paper: "This is due to different seasons and study regions between our and their studies. The summer boundary layer in the ENA region is deeper and less stable with higher cloud tops (e.g., Klein and Hartmann, 1993; Ding et al., 2021; King et al., 2013) compared to the NE Pacific in Zhang et al. (2022) and the NE Atlantic region in Zhang and Feingold (2023)." (Lines 750-753)

**References**

Arola, A., Lipponen, A., Kolmonen, P., Virtanen, T. H., Bellouin, N., Grosvenor, D. P., Gryspeerdt, E., Quaas, J., & Kokkola, H. (2022). Aerosol effects on clouds are concealed by natural cloud heterogeneity and satellite retrieval errors. Nature Communica)ons, 13(1), 7357. hrps://doi.org/10.1038/s41467-022-34948-5Bennartz, R.: Global assessment of marine boundary layer cloud droplet number concentration from satellite, J. Geophys. Res.,112, D02201, doi:10.1029/2006JD007547, 2007.

Chen, Y.-C., Christensen, M., Stephens, G. L., and Seinfeld, J. H.: Satellite-based estimate of global aerosol–cloud radiative forcing by marine warm clouds, Nature Geosci., 7, 643–646, https://doi.org/10.1038/ngeo2214, 2014.

Christensen, M. W., Jones, W. K., and Stier, P.: Aerosols enhance cloud lifetime and brightness along the stratus-tocumulus transition, P. Natl. Acad. Sci. USA, 117, 17591–17598, https://doi.org/10.1073/pnas.1921231117, 2020.

Ding, F., Iredell, L., Theobald, M., Wei, J., & Meyer, D. :. PBL Height From AIRS, GPS RO, and MERRA-2 Products in NASA GES DISC and Their 10-Year Seasonal Mean Intercomparison. *Earth and Space Science*, *8*(9). https://doi.org/10.1029/2021ea001859, 2021

Feingold, G., Goren, T., & Yamaguchi, T. (2022). Quantifying albedo susceptibility biases in shallow clouds. Atmospheric Chemistry and Physics, 22(5), 3303–3319. https://doi.org/10.5194/acp-22-3303-2022.

King, M. D., Platnick, S., Menzel, W. P., Ackerman, S. A., & Hubanks, P. A.:  Spatial and Temporal Distribution of Clouds Observed by MODIS Onboard the Terra and Aqua Satellites. *IEEE Transactions on Geoscience and Remote Sensing*, *51*(7), 3826-3852. https://doi.org/10.1109/tgrs.2012.2227333, 2013

Toll, V., Christensen, M., Quaas, J., and Bellouin, N.: Weak average liquid-cloud-water response to anthropogenic aerosols, Nature, 572, 51–55, https://doi.org/10.1038/s41586-019-1423-9, 2019.

Xue, H. and Feingold, G.: Large-Eddy Simulations of Trade Wind Cumuli: Investigation of Aerosol Indirect Effects, J. Atmos. Sci., 63, 1605–1622, https://doi.org/10.1175/JAS3706.1, 2006.

Zhang, J., Zhou, X., Goren, T., and Feingold, G.: Albedo susceptibility of northeastern Pacific stratocumulus: the role of covarying meteorological conditions, Atmos. Chem. Phys., 22, 861–880, https://doi.org/10.5194/acp-22-861-2022, 2022.

Zhang, J., and Feingold, G.: Distinct regional meteorological influences on low-cloud albedo susceptibility over global marine stratocumulus regions, Atmos. Chem. Phys., 23, 1073–1090, https://doi.org/10.5194/acp-23-1073-2023, 2023.

Zhou, X., & Feingold, G. (2023). Impacts of mesoscale cloud organization on aerosol-induced cloud water adjustment and cloud brightness. Geophysical Research Letters, 50, e2023GL103417. https://doi.org/10.1029/2023GL103417

---

## Author Comment (AC2)

**Responses to reviewer comments**
The original comments are in *blue italic font*, and our response are in black font.

**Reviewer #2:**
*This study provides a comprehensive look at daytime evolution of low cloud susceptibility to droplet number (Nd) perturbations. These susceptibilities are derived from snapshots of cloud and radiation fields taken from passive sensors (SEVIRI), onboard a geostationary satellite (Meteosat11), within a 10-by-10-degree box over the NE Atlantic region. The authors found persistent "U-shape" evolutions in cloud susceptibility, including cloud water, cloud albedo, and cloud fraction, from which they argue that polar-orbiting satellite derived cloud susceptibility underestimate the daytime-mean cloud susceptibility. They attribute the observed evolution in cloud susceptibility to two main hypotheses that they formulated and tested: i) clouds have "memory" of their past susceptibility signal associated with the cloud state they transition from; ii) cloud responses to Nd perturbations have a longer timescale than the satellite observing time interval (30 mins).*

*Overall, I found the study interesting and intriguing in some sense. Clearly, the authors have put in lots of thoughts on this problem and lots of efforts in interpreting these results. The manuscript is nicely constructed and organized. That being said, I do think there are some key/fundamental issues that need to be addressed and justified first. If these concerns and issues can be sufficiently addressed, I believe this work has the potential to provide new, impactful insights and make significant contribution to the field.*

We are grateful to the reviewer for the thorough review of our manuscript and for providing constructive comments. Below, we provide a point-by-point response to each comment and details of the modifications made to the text and figures to address these points.

*Major concerns/comments:*
*1. The authors adopted a methodology for deriving cloud susceptibility that uses spatial regression of cloud properties to Nd within satellite snapshots. This is done to minimize the impact from confounding meteorology. I found the extension of this approach to cloud fraction susceptibility is a bit unjustified, and I'm in general concerned about the appropriateness of applying this methodology to cloud fraction susceptibility. Cloud fraction is a tricky quantity by definition. It depends strongly on the spatiotemporal scale of one's investigation, for example, the distribution of daily cloud fraction will look very different from that of monthly cloud fraction at any given location on Earth. The degree of spatial aggregation also strongly affects cloud fraction, e.g., cloud fraction reduces to binary (zero and one) at pixel level. Overall, I find the discussion/explanations around CF susceptibility quite vague and often led to confusion, which could potentially be related to an ill-defined methodology. Some of my specific concerns are:*

*1a. How confident are you that, at 25-km resolution, variations in CF is not simply coming from the geometry of the cloud field, such that one scene captures the majority of a cloud while another scene captures only a small portion of the same cloud, resulting in a large difference in cloud fraction between the two scenes, while Nd has nothing to do with it.*

   Thank you for raising this important point. Most of the concern pertains to stratiform clouds with an area larger than the 1-degree scene, where variations in the $0.25° \times 0.25°$ mean

CF could result from sampling cloud edges or the cloud center. To assess the influence of cloud geometry on the retrieved CF susceptibility, we excluded $1° \times 1°$ scenes with a range of $0.25°$ CF greater than 90%, a range of $N_d$ smaller than 60 $cm^{-3}$, and the total cloud areas larger than 20,000 km². Cloud area is defined for each cloud object by the number of contiguous cloudy pixels. A total of ~17,000 scenes were removed, which accounts for ~24% of the samples. As shown in Figures R1-R4, compared with the original results, removing the $1° \times 1°$ scenes that possibly sampled different parts of the cloud with small variation in $N_d$ do not alter the conclusions of this study. Since cloud geometry exhibited minimal impact on the overall CF susceptibility, we opted to retain these scenes with large CF variations to maintain consistency with other cloud susceptibilities. A discussion regarding this concern has been incorporated into the methodology section.

"Due to the highly variable nature of CF, variation in the $0.25°$ CF may result from quantifying edges or centers of the same cloud layer instead of from $N_d$ perturbations. To test the influence of cloud geometry on the retrieved CF susceptibility, we removed any $1° \times 1°$ grid box with variation in the $0.25°$ CF greater than 0.9 while the variation in the $0.25°$ $N_d$ less than 60 $cm^{-3}$, and $0.25°$ CF in the $1°$ box sample the same cloud. A total of 17,000 scenes were removed, which accounts for ~24% of the total samples. Removing these scenes does not change the conclusions of CF susceptibility in this study (not shown), which demonstrates that cloud geometry has minimal impact on the retrieved CF susceptibility." (Lines 215-221)

[Figure]

Figure R1. Same as Figure 2, but excluding scenes with large variation in $0.25°$ CF and small variation in $0.25°$ $N_d$.

[Figure]

Figure R2. Same as Figure 3, but excluding scenes with large variation in 0.25° CF and small variation in 0.25° $N_d$.

[Figure]

Figure R3. Same as Figure 5, but excluding scenes with large variation in 0.25° CF and small variation in 0.25° $N_d$.

[Figure]

Figure R4. Same as Figure 11, but excluding scenes with large variation in 0.25° CF and small variation in 0.25° $N_d$.

*b. On lines 335-336, the readers are directed to the next section for explanations on the CF susceptibility pattern seen in Fig. 2c. In the next section, your discussion and hypotheses testing are targeted at CF susceptibility evolutions, while I'm still missing an explanation for the non-precip thin cloud that exhibit potentials in cloud fraction enhancement.*

As we explained in response to a relevant comment from Reviewer #1, increases in aerosol decrease cloud drop size and increase cloud drop number concentration near cloud top, which enhance the cloud top radiative cooling rate and the entrainment rate. The enhanced radiative cooling and entrainment induce downdraft and mixing from cloud top (e.g., Xue and Feingold, 2006). These factors contribute to destabilize the boundary layer and facilitate moisture transport from the ocean surface to clouds, thus enhance new cloud formation and extend cloud lifetime (e.g., Christensen et al. 2020). This hypothesis is consistent with and supported by the relative low CF for these clouds (Fig. S2a) and the diurnal variation in LWP susceptibility for non-precipitating thin clouds (Figure 4c). In the morning, the boundary layer is typically shallower and well-mixed with clouds coupled to the surface. Therefore, the CF susceptibility for thin clouds is large positive in the morning, and gradually decreases from morning to noon. However, the near zero CF susceptibility in the afternoon is not supported by this hypothesis. Further analyses and model simulations are needed to better understand the diurnal evolution of aerosols' impact on entrainment rate, boundary layer state, cloud cover and

lifetime to explain the observed daytime variation of CF susceptibility for non-precipitating thin clouds. The opposite signs observed in LWP and CF susceptibility suggest that the aerosol indirect effect likely redistributes cloud water horizontally, causing clouds to thin and widen. A related discussion has been added to Figure 2 on the daytime mean CF susceptibility and to Figure 5 on the daytime variation of CF susceptibility.

"A possible explanation for the increased CF is the enhanced cloud top radiative cooling rate from aerosol perturbations help to mix the boundary layer, facilitate moisture transport from the ocean surface to cloud, and therefore favor new cloud formation and extend cloud lifetime (e.g., Christensen et al. 2020). This hypothesis is consistent with and supported by the relative low CF for these clouds (Fig. S2a) and the diurnal variation in LWP susceptibility for non-precipitating thin clouds, which will be discussed in the next section. The opposite signs of LWP and CF susceptibilities indicate that the AIE might redistribute cloud water horizontally and make the thin clouds thinner and wider." (Lines 406-411).

"Another possible explanation on the evolution of CF susceptibility is the influence of aerosols on boundary layer mixing and the evolution of boundary layer from morning to noon. The enhanced entrainment rate and radiative cooling rate from $N_d$ perturbations help to destabilize the boundary layer and transport moisture from the ocean surface to clouds, which facilitate new cloud formation. As the boundary layer is typically well mixed in the morning with clouds coupled to the surface, this impact is strongest in the morning and gradually decrease from morning to noon. In the afternoon, on the other hand, thin clouds transition from all three states have near-zero CF responses to $N_d$ perturbations, which cannot be explained by the hypothesis above. Further analyses and model simulations are needed to better understand the diurnal evolution of aerosols' impact on entrainment rate, boundary layer state, cloud cover and lifetime to explain the observed daytime variation of CF susceptibility for non-precipitating thin clouds." (Lines 551-560)

*c. Fig. 11 is another example that I am having difficult times wrapping my head around. While Fig. 11a and b suggest that the frequency of occurrence of cloud state in the LWP-Nd space is indeed changing throughout the day, the null contribution from shown in Fig. 11c troubles me a bit. I guess it's related to the pattern we see in Fig. 2c, which I still struggle to decipher.*

In Figure 11, red lines represent the daytime variation contributed from the occurrence frequency of different cloud states (estimated by weighting the daytime mean cloud susceptibility with the half-hourly frequency of occurrence of clouds in the LWP-$N_d$ parameter space), while blue lines represent the contribution from the cloud susceptibilities for different cloud states (estimated by weighting the half-hourly cloud susceptibility in the LWP-$N_d$ parameter space with the daytime mean frequency of occurrence of clouds). As seen in Figure 6, the daytime variation in the occurrence frequency of non-precipitating thick clouds is minimal, so that red lines in Figure 11 are mainly contributed from variations in the occurrence frequency of precipitating and non-precipitating thin clouds. As shown in Figure 2, non-precipitating thin clouds are more susceptible than precipitating clouds in LWP, with daytime mean LWP susceptibilities of -0.7 vs. -0.4. Meanwhile, the occurrence frequency of non-precipitating thin clouds reaches a daily maximum in the afternoon and a daily minimum in the evening, while the occurrence frequency of precipitating clouds is highest in the evening (Figures 4 and 9). Therefore, the red line in Figure 11a is most negative in the afternoon and less negative in the evening. The same pattern is observed for albedo susceptibility in Figure 11b. On the other hand,

the daytime mean CF susceptibility for non-precipitating thin clouds and precipitating clouds are very similar at 0.08 and 0.09, respectively (Figure 2c). Therefore, contributions from the changes in occurrence frequency are muted by the similar CF susceptibilities of these two clouds in Figure 11c.

*2: Regarding the explanation for the "U-shape" evolutions, it seems a bit suspicious to me that all 3 quantities show the same shape of evolution. Moreover, you use cloud transition categories to explain the observed U-shape evolution, yet you see the similar "U-shape" for individual categories, suggesting to me that there are something fundamental that has not been teased out. This makes wonder if this is due to retrieval artifacts associated with relatively high solar zenith angle during early morning and late afternoon? Smally and Lebsock (2023) made a comparison between retrieved LWP from tau & re using adiabatic assumption with microwave imager retrieved LWP, onboard geostationary satellites. They found non-negligible bias even at intermediate SZA. The same bias applies for Nd retrievals as well, in a more pronounced way I suppose, given the high order dependence of Nd calculation on re. I wonder if this "U-shape" evolution can be explained by tau & re retrieval biases as a function of SZA. To me, even small biases can produce the trend you see towards higher SZAs (i.e. towards early morning and late afternoon). I suppose one can play with some synthetic data to rule this possibility out.*

Thank you for this very good question. In the study by Smally and Lebsock (2023), it was found that the ABI retrieved LWP from GOES-16 and GOES-17 has larger uncertainty at solar zenith angle (SZA) > 60-65° and viewing zenith angle (VZA) > 65°. The VZA at Azores is ~50°, which is below the VZA threshold found in Smally and Lebsock (2023). To minimize the retrieval uncertainty in $r_e$, $\tau$, and $N_d$, we carefully screened the pixel-level Meteosat retrieval with $r_e$>3$\mu m$, $\tau$>3, SZA<65°, and further removed cloudy pixels at the cloud edge.

To demonstrate that the observed "U-shaped" diurnal variation of cloud susceptibilities is unlikely due to the systematic bias of $r_e$ and $\tau$ retrievals at large SZA, we testify from the following two aspects with additional analyses.

Firstly, if the daytime variation is driven by retrieval bias at large SZA, we would expect the susceptibility exhibiting a symmetric pattern at local noon. However, as seen in Figure R5 (same as Figures 4, 6, 9 in the paper but with three cloud states combined), only the non-precipitating thin clouds show symmetric patterns in LWP and albedo susceptibilities. The LWP susceptibilities for precipitating clouds and non-precipitating thick clouds, as well as the CF susceptibilities for all three cloud states exhibit asymmetric pattern at local noon.

Secondly, we did a sensitivity test focusing on overcast clouds only, with the mean cloud fraction in the 1° × 1° scenes larger than 0.9. At large SZA, the uncertainty of the Meteosat cloud microphysics retrievals should be smaller for stratiform clouds compare to cumulus clouds due to the plane-parallel assumption and cloud 3-D effect. Therefore, if the daytime variation in cloud susceptibilities is driven by the retrieval bias, we would expect less daytime variation in cloud susceptibilities for overcast clouds, especially in the early morning and in the evening. Contrary to this expectation, as shown in Figure R6, overcast clouds exhibit stronger and asymmetric daytime variation of LWP and albedo susceptibilities than that for all the boundary-layer clouds shown in Figure 3. For example, the magnitude of change in LWP susceptibility for overcast clouds is 1.0 compared to 0.4 for all clouds.

Based on these two arguments, the observed "U-shaped" diurnal variation of cloud susceptibilities is unlikely due to the systematic bias in $r_e$ and $\tau$ retrievals at large SZA. We added the following discussion to the paper regarding to this comment: "In addition, the

asymmetric evolution patterns of LWP, $\alpha_c$, and CF susceptibility for non-precipitating thick and precipitating clouds indicate that the daytime variation of cloud susceptibility is not due to retrieval uncertainties related to SZA. To minimize the possible retrieval uncertainties related to large SZA, all Meteosat pixel-level cloud retrievals are filtered with SZA<65°." (Lines 701-704)

[Figure]

Figure R5: Daytime variation of cloud susceptibilities for three cloud states. (a) percentage of occurrence of different cloud states, (b) cloud LWP susceptibility ($dln(LWP)/dln(N_d)$), (c) cloud albedo susceptibility ($d\alpha_c/dln(N_d)$), and (d) cloud fraction susceptibility ($dCF/dln(N_d)$) for non-precipitating thin clouds. The blue lines are for precipitating clouds, the orange lines are for non-precipitating thick clouds, and the green lines are for non-precipitating thin clouds.

[Figure]

Figure R6. Same as Figure 3, but for overcast clouds with the 1° × 1° mean cloud fraction larger than 0.9.

*3. Assuming daytime evolution in cloud susceptibility is not SZA-dependent for now, an alternative explanation for the "U-shape" evolution is the change in meteorology, i.e., daytime evolution in boundary layer (BL) depth. From morning to early afternoon, BL deepens and becomes decoupled, enhancing the role of entrainment feedback in governing cloud susceptibility. In the afternoon, BL re-couples. This could possibly give you the "U-shape" evolution in cloud susceptibility. I wonder if the authors have considered this as an alternative explanation and tested it?*

Thank you for providing this valuable aspect on the potential mechanism behind the "U-shape" evolution of cloud susceptibilities. To explore the influence of the boundary layer state on cloud susceptibilities, we tested using the Meteosat retrieved cloud top height and cloud base height as indicators of boundary layer depth and coupling states over the ENA region. As shown in Figures R7 and R8, the Meteosat-retrieved cloud top height and base height reach the daytime minimum between 14-16 LST, raises in the late afternoon and reaches the daytime maximum in the evening.

Therefore, the Meteosat retrieved cloud top or cloud base height likely cannot represent the boundary layer depth or cloud coupling state in the ENA region in summer. Unfortunately, there is no better indicator of cloud coupling state from the Meteosat retrievals for us to test the

influence of cloud coupling to sea surface on cloud susceptibilities. Consequently, we added a discussion on the daytime evolution of boundary layer to explain the evolution of cloud susceptibility in the paper.

"In the early morning (e.g., 9-10 LST), with higher frequency of occurrence of thick clouds transition to thin clouds, the LWP susceptibility for the thick-to-thin category is less negative compared to later time (dash line with diamond symbols in Figures 5a, b). This is likely due to the less negative LWP susceptibility for non-precipitating thick clouds in earlier time (e.g., 7-9 LST, not shown), which is because boundary layer clouds are coupled to the sea surface at night and in the early morning transporting moisture from ocean to clouds and compensate the moisture loss from aerosol enhanced entrainment (e.g., Sandu et al., 2008)." (Lines 536-541).

[Figure]

Figure R7. Daytime variation of the Meteosat retrieved cloud top height for different cloud states. The blue line is for precipitating clouds, the orange line is for non-precipitating thick clouds, the green line is for non-precipitating thin clouds, and the black line is for all warm boundary layer clouds.

[Figure]

Figure R8. Same as Fig. R7, but for Meteosat retrieved cloud base height.

*4. Regarding the "memory" hypothesis, I have to say that I'm quite lost in this argument, by the design of the analysis, by the inconsistency that I spotted when reasoning with this argument, and by the actual physical meaning of "memory" in the context of cloud susceptibility.*
*a. Fundamentally, the concept of "memory" requires a Lagrangian perspective, however, I believe all the analyses done in this study are based on a Eulerian framework. This means that the "memory" in this study is actually the domain "memory" of the clouds that were previously present in the domain, not cloud "memory" of its past states. I believe one needs to at least do a grid-box tracking, if not individual cloud tracking, to support this argument, or show that clouds are stationary/semi-stationary in the study region.*

[Figure]

Figure R9: Probability density function (PDF, black lines) and cumulative density function (CDF, blue lines) of the 1000 to 900 hPa mean wind speed (solid lines) and the wind speed at 1000 hPa (dash lines) from the ERA5 reanalysis during the study period (July, 2018-2021).

Thank you for this constructive comment. We assessed the low-level main flow in the study region during the study period. The mean wind speed for each 1° × 1° grid box is averaged from the 0.25-degree ERA5 data based on the central longitude and latitude of each box, and the PDF and CDF of the PBL mean wind speed for all the grid boxes in the study domain during the study period is shown in Figure R9. During the summer in the study region, low wind conditions prevail in the boundary layer, with the mean wind speed being less than 10m/s for 85% of the time and less than 7m/s for 60% of the time. Since the cloud state used in this study is defined using the mean effective radius and liquid water path for each 1°×1° grid box, clouds with mean wind speed weaker than 7m/s (60% of data) move less than 50 km within the two-hour tracking period, thus the cloud state still captures the majority of the original clouds within the same grid box.

To support our hypothesis that the evolution of cloud susceptibility is contributed by the transition of cloud state combined with cloud memory, instead of changing of cloud state from advection, we reproduce Figure 5, 7, 9 in the paper under low wind speed conditions (wind speed < 7m/s); results for non-precipitating thin clouds are shown in Figure R10. As seen in Figure R10a, the occurrence frequency of non-precipitating thin clouds transition from other three cloud states during low wind speed days exhibit similar values and daytime variation as under all wind

conditions. Additionally, under low wind speed conditions, non-precipitating thin clouds transitioned from thick clouds previously exhibit consistent more negative LWP and albedo susceptibilities than thin clouds remained thin (Figure R10b, c). Difference between the two transition groups under low wind speed conditions is slightly larger than that under all wind speed conditions.

[Figure]

Figure R10. Daytime variation of non-precipitating thin clouds transition from non-precipitating thin clouds (thin → thin, solid line with circle symbols), precipitating clouds (rain → thin, solid line with triangle symbols), and non-precipitating thick clouds (thick → thin, dash line with diamond symbols) in previous two hours with PBL mean wind speed less than 7m/s. Symbols for different state transitions are noted in (b). In (b)-(d), filled markers indicate data points that are significantly different from the other two groups (p<0.05), while open markers indicate statistical insignificance.

In summary, clouds are mostly stationary in the study period and changes in cloud states in a fixed grid-box likely represent the transition of cloud state as well as its impact on cloud susceptibility from the transition. We have revised the manuscript and added a discussion on the credibility of our tracking method to the methodology section and to Fig. 5 as follow.

"To investigate the dependences of AIE on previous cloud states and quantify the influence of cloud memory on the estimated cloud susceptibility, we track the historical cloud state over a 1° × 1° grid box for a two-hour period. During the summer in the study region, low wind conditions prevail in the boundary layer, with the mean wind speed being less than 10m/s for

85% of the time and less than 7m/s for 60% of the time (not shown). Therefore, in most cases, less than half of the clouds exit the grid box within the two hours, allowing us to track the previous cloud state within the same grid box. The influence of cloud memory is assessed by comparing the cloud susceptibilities of clouds that undergo a transition in cloud state with those that do not experience such a transition. Section 3.4 includes more details and discussions on the sensitivity of tracking time and the influence of advection on our classification." (Lines 256-264)

"To further isolate the influence of cloud advection from the tracked transition of cloud state and cloud susceptibilities, we focus on low wind conditions with wind speed less than 7m/s when clouds are stationary during the two-hour period (60% of time). The influence of cloud state transition on cloud susceptibility is stronger with larger differences in LWP and $\alpha_c$ susceptibilities between clouds experienced transition in cloud state with clouds did not (not shown). Therefore, our tracking method can capture the signal of cloud state transition and its impact on cloud susceptibilities during summer in this study region." (Lines 570-576)

*b. One example of the inconsistency that I observed is that in Fig. 5, if I follow the argument correctly, thick→thin transition will lead to more negative LWP susceptibility based on the LWP-Nd plot, however, at early morning where thick→thin is the most frequently occurring type, the overall LWP susceptibility is the highest (Fig. 4b). Furthermore, how do you explain the "U-shape" in the evolution that is specific to thick→thin transition clouds, they all have the same type of "memory" according to your argument.*

Based on our hypothesis, the current cloud susceptibility is influenced by both the cloud state and the cloud susceptibility at the previous time. For example, the susceptibility of thin clouds transitioning from thick clouds at 9LST depends on the susceptibility of the original thick clouds in the preceding hours. As seen in Figure R11, thick clouds are less susceptible in the early morning (7-9 LST), resulting in a less negative susceptibility for the thick→thin clouds at later time (dash line with diamond symbols in Figure 5b in the paper), even though such transitions are most frequent in the morning.

Not only the thick→thin cloud category, but also the thin→thin, the thin→thick, and the thick→thick categories exhibit "U-shaped" evolution. For example, the thin→thin cloud category exhibit a less negative LWP susceptibility from 9-11LST and it decreases to more negative from 11-14 LST (a "U-shaped" daytime evolution), which is consistent with the diurnal variation for non-precipitating thin clouds from 7-12 LST shown in Figure R11. The thin→thick cloud category exhibits a similar less negative LWP susceptibility from 9-11 LST, and it decreases to more negative from 11-14 LST (Figure 7b). The LWP susceptibility for the thick→thick cloud category decreases from around –0.8 at 9LST to –1.3 at noon (Figure 7b). The only cloud transition category that cannot be explained by our hypothesis is the rain→thick, where the LWP susceptibility is more negative in early morning. The following discussion has been added to Fig. 5 for clarity.

"In the early morning (e.g., 9-10 LST), with higher frequency of occurrence of thick clouds transition to thin clouds, the LWP susceptibility for the thick-to-thin category is less negative compared to later time (dash line with diamond symbols in Figures 5a, b). This is likely due to the less negative LWP susceptibility for non-precipitating thick clouds in earlier time (e.g., 7-9 LST, not shown), which is because boundary layer clouds are coupled to the sea surface at night and in the early morning transporting moisture from ocean to clouds and compensate the moisture loss from aerosol enhanced entrainment (e.g., Sandu et al., 2008).

Similarly, as shown in Figure 5b, differences between the thin-to-thin and thick-to-thin categories are less pronounced in the afternoon when the LWP susceptibilities of thin and thick clouds are close to each other (Figures 4b and 6b.)" (Lines 535-542).

[Figure]

Figure R11. Same as Figure R5, except from 7 to 18 LST. Daytime variation of cloud susceptibilities for different cloud states. (a) Percentage of occurrence of different cloud states, (b) cloud LWP susceptibility ($dln(LWP)/dln(N_d)$), (c) cloud albedo susceptibility ($d\alpha_c/dln(N_d)$), and (d) cloud fraction susceptibility ($dCF/dln(N_d)$). Different colors represent different cloud states: blue for precipitating clouds, orange for non-precipitating thick clouds, and green for non-precipitating thin clouds.

*c. I wonder at what timescale do you expect these clouds to detach/decorrelate from their "memory" of past states? Existing literature seems to suggest a decorrelation timescale that is much shorter than your 2-hr window, on the order of 10 to 15 mins, for marine boundary layer clouds. I wonder if you could elaborate on how does this decorrelation timescale fit in your 2-hr "memory" hypothesis?*

Our hypothesis, suggesting that clouds retain the memory of ACI from their previous states, is primarily based on the timescale of cloud responses to aerosol and $N_d$ perturbations. We believe this timescale is a different concept compared to the detachment timescale mentioned by the reviewer. For a single cloud object, its timescale is on the order of 10 to 15 mins. For cloud

systems, the decorrelation, formation, and development timescale for the system is on the order of ~20 hours for cloud fraction, and ~15 hours for cloud LWP (e.g., Eastman et al., 2016; Christensen et al., 2020). In our study, we are interested in the mean cloud state for a $1° \times 1°$ grid box, and the two-hour tracking window is much shorter than the timescale of the system. Additionally, for the timescale of cloud response to $N_d$ perturbation, previous numerical simulations and satellite observations found that the LWP response is time sensitive, and its sensitivity increases with time. The LWP response takes ~4-5 hours to manifest and ~20 hours to reach the equilibrium for both precipitating (e.g., Gryspreedt et al., 2021) and non-precipitating clouds (e.g., Gryspreedt et al., 2021; Glassmier et al., 2021). Since the $N_d$-LWP sensitivity takes time to reach an equilibrium, we hypothesize that if the clouds undergo a state change during this time, it retains the LWP responses of its previous state.

*d. I'm familiar with the concept that cloud have "memory" of their near-past state, in which clouds carry-over their past tendencies for a short period of time. However, I am afraid that I don't quite follow the argument that clouds have a "memory" of its past 'susceptibility.' I wonder if you could elaborate more on the physical mechanism of it. I believe the schematic (Fig. 8) you shown refers to the concept I mentioned, such that stronger evaporation & entrainment are carried over if the cloud just transitioned from thick to thin. It speaks to the temporal evolution of clouds, but how does this work for a "memory" of susceptibility, which is essentially a spatial regression, is less clear to me.*

       Thank you for raising this insightful question. The LWP susceptibility, which is a measure of rate of change in LWP with increasing $N_d$, is mainly driven by evaporation rate and entrainment rate at cloud top. The possible physical processes and mechanisms for clouds sustaining the memory of its LWP susceptibility is that: for an initially thick clouds with large LWP, the larger cloud top radiative cooling rate will result in a larger entrainment rate and evaporation rate near cloud top. The increased evaporative cooling from higher evaporation rate further enhances entrainment rate. The larger cloud top radiative cooling and entrainment rate also induce downdraft and mixing from cloud top, which in turn enhance entrainment rate. All of the above factors form a positive feedback loop and set up an environment favor for stronger entrainment and evaporation. These feedback and environment will not change immediately even when the cloud LWP decrease and cloud transition to a thin state. This hypothesis is supported by model simulation where they found the maximum cloud top radiative cooling rate is more related to cloud LWP and the enhanced evaporation rate from increasing aerosols is driven by the cloud top radiative cooling rate rather than by the faster evaporation from smaller droplets (Williams and Igel, 2021). Regarding the comment on the calculated susceptibility is the spatial regression of the $N_d$-LWP relationship, I think this quantification is based on the hypothesis that the temporal change in LWP from $N_d$ perturbations can be represented by the spatial correlation.

*Minor comments:*
- *Line 36-37, 600, 669-671, the authors made the point that susceptibility derived from polar-orbiting satellites, Aqua in particular, underestimates the daily mean value of cloud susceptibly. While this is true based on the findings from this study, I would like to raise the point that this underestimation does not necessarily translate into an underestimation of the actual cloud response, as the simple arithmetic mean of susceptibility ('local derivatives') is*

*not the same as a time integral of cloud responses, which is more relevant to the scaling up of radiative effects of ACI.*

Thank you for the clarification. We agree that our method estimate the cloud susceptibility instead of the integrated cloud responses over time. In all these places, we carefully used the words of estimated cloud susceptibility instead of cloud responses. We added the following discussion to section 5. "Notice that both the daytime variation and the daytime mean values of cloud susceptibilities in this study are estimated based on the regression analysis on spatial data within each satellite time step with the assumption that the temporal change of cloud properties from $N_d$ perturbations can be represented by the spatial relationships." (Lines 814-817).

- *Line 48, I guess you need more reference here than just Albrecht 1989, as it only speaks to the precip-suppression effect.*

  We added more references here: e.g., Albrecht, 1989; Xue and Feingold, 2006; Chen et al., 2014; Gryspeerdt et al., 2019.

- *Line 109, note that Gryspeerdt et al. (2021) did not use geostationary satellite but rather polar-orbiting satellites, i.e., Terra and Aqua.*

  This sentence and paragraph have been modified. Thanks.

  "Observational and modeling studies have shown that aerosol-cloud interaction processes take hours to reach the equilibrium state and the sensitivity of AIE is time dependent. For instance, Glassmeier et al. (2021) used a Gaussian-process emulation and derived the adjustment equilibration timescale for LWP to be ~20 hours. By tracking the ship tracks in satellite observations, Gryspperd et al. (2021) found a similar timescale of AIE of ~20 hours or longer and the magnitude of LWP susceptibility increases with time. In addition, Christensen et al. (2020) discovered that influence of aerosols on cloud LWP, CF, and cloud top height persists two to three days by tracking cloud systems in satellite observations. To conclude, the sensitivity of cloud responses to $N_d$ perturbations changes with time, the assumption that AIE has no memory of its past state likely fails." (Lines 118-126)

- *Line 114, I thought you did not "track" any cloud or cloud field and used with the Eulerian framework, I would be more precise here to avoid confusion.*

  Changed.

- *Line 137, reference for LWP calculation formular?*

  Added reference here (Minnis et al. 2011, 2020).

- *Line 155, cw is a function of pressure as well.*

  Modified.

- *Line 172-173, did you average tau and re first and then calculate Nd or calculate Nd first then average? This difference can induce huge bias in albedo susceptibility quantification as shown in Feingold et al. (2022, ACP).*

  Thank you for the clarification and the reference. In this study, we first calculated the pixel-level $N_d$ based on pixel-level $r_e$ and $\tau$ using Eq. (1), then all the pixel-level cloud retrievals are averaged to 0.25-degree to smooth out noise and uncertainties from homogeneity and spatial covariation. Our averaging method is consistent with the recommendation in Feingold et al. (2022). We have modified these lines in the method section.

  "As found by Arola et al. (2022) and Zhou and Feingold (2023), the retrieved cloud susceptibilities are sensitive to small-scale cloud heterogeneity, the co-variability between cloud properties and $N_d$, and the spatial scale of cloud organization. To reduce the biases resulting from heterogeneity and co-variability, we first average the 3-km pixel-level cloud

retrievals to a regular $0.25° \times 0.25°$ grid for each half-hourly time step. As suggested by Feingold et al (2022), $N_d$ retrieval was performed at pixel-level using Eq. (1), and then averaged to a $0.25°$ resolution." (Lines 194-198).

- *Line 175, note Zhou et al. (2021) used 2x2 grid box, if I recall correctly, worth double checking.*
  Yes, you are right. They used a $2° \times 2°$ grid box. The reference has been modified.
- *Line 187, why cloudy pixels at cloud edge are set as clear?*
  At cloud edge, the adiabatic assumption for the $N_d$ retrieval is more likely to fail so that the retrieved $N_d$ suffers larger uncertainty. As the $N_d$ for cloudy pixels at cloud edge are removed, the CF is set as clear for those pixels for consistency in CF susceptibility. Similarly, LWP is set as NAN for those pixels.
- *Line 218-219, why you opt for a Eulerian framework here? Isn't a Lagrangian framework the appropriate choice here as you want to investigate cloud "memory"?*
  As discussed in a previous comment, our analyses indicate that boundary layer clouds are mostly stationary in the study region. To assess the potential influence of cloud advection on our results, we specifically examined low wind speed conditions. Based on results shown in Figures R9 and R10, the Eulerian framework can effectively capture the signal of cloud state transition as well as its impact on cloud susceptibilities. Relevant revisions have been made here and later in Fig. 5 in the paper.
- *Line 254, what do you mean by "witnessed by clouds"?*
  This sentence has been modified. "The absence of a precipitation suppression signal is likely attributed to the relatively modest precipitation occurred in this region during summer (e.g., Wu et al., 2020; Zheng and Miller, 2022), where the effect of precipitation suppression is minimal and the entrainment drying effect dominates." (Lines 296-297).
- *Line 266, I wouldn't use "hypothesis" here as these entrainment feedback mechanisms have been previously established, using large-eddy simulations.*
  Changed.
- *Line 280-281, what do you mean by "meteorological influences on clouds likely dampen the signal of the AIE…" I feel like you try to say meteorological confounding tends to obscure the AIE signal, probably need to rephrase here.*
  Changed accordingly. Thanks for the suggestion.
- *Line 352-353, I don't think you have diurnal cycles, only daytime, right?*
  Yes, we changed the term "diurnal" to "daytime" throughout the paper.
- *Fig. 3c, the day-to-day (or spatial variation within the 10 by 10 box) variation is huge, at a given time of the day, is it possible to tell the sign of CF susceptibility with statistical significance?*
  The switch in sign for albedo susceptibility is statistically significant at a 95% confidence level based on a student's t-test. However, the switch in sign for CF susceptibility is not. We add discussion on the significance of sign of CF and albedo susceptibilities. "The switch in sign for albedo susceptibility is statistically significant at a 95% confidence level, while the switch in sign for CF susceptibility is not statistically significant." (Lines 450-451).
- *Line 439-440 (Figure 5b), I don't follow this argument, your susceptibility for each individual transition group shouldn't be affect by their relative frequency of occurrence at a given time, if I understand your method correctly, then why fewer samples in the thick/precip→thin groups lead to less difference between the groups?*

Agree. The original sentence was not accurate enough. The difference between the rain->thin and other two groups are still pronounced and significant in the local afternoon. However, the difference between the thin->thin and thick->thin category is not significant in the afternoon, which is likely due to the similar LWP susceptibilities for thin and thick clouds as shown in Figure R10b. This sentence has been deleted.

- *Line 454-456, again, why not using Lagrangian tracking in this assessment?*
  Please refer to the previous response.

- *Table. 1, I wonder if there are more effective ways of illustrating hypotheses than showing them in a table, the current structure of the table is quite difficult to decipher, at least for me. In particular the N/A boxes confuse me, and I don't see discussion of them in the main text.*
  We have changed the layout of Table 1. The new table is inserted at Line 475.

- *Line 478-481, these arguments are vague, if your "memory" argument stands, shouldn't we see a lagged evolution in thin clouds compared to thick clouds, instead of similar evolution?*
  Agree, the original sentence was not accurate enough. We checked the lagged correlation between cloud susceptibilities of thick clouds and cloud susceptibilities for thin clouds transitioned from thick. For all three cloud variables, the lagged correlations reach the maximum of ~0.75 at the fourth time step, which equals two hours. This sentence has been modified to add the lagged effect.

- *Line 488, why this is particularly prominent in the morning, I could not think of a reason following your "memory" argument.*
  Agree. Similar as the previous response, the difference between the two categories is likely contributed by the evolution of LWP susceptibilities for the thin and thick clouds during the day, rather than by variations in sample numbers. This sentence has been modified as follow:
  "These differences are particularly prominent from late morning to noon and become not significant in the afternoon. As discussed before, difference between the thick-to-thick and the thin-to-thick categories are due to the LWP susceptibilities for thick and thin clouds of previous time, while the smaller differences in the early morning and afternoon could be attributed to cloud coupling states at these times." (Lines 606-610)

- *Line 567, worth repeating the question here.*
  Done.

- *Fig. 10b, why the difference between thin→rain and thick→rain is reversed during midday, compared to early morning?*
  Good question. Not only the thick-to-precipitating, but also the thin-to-precipitating categories show much less negative LWP susceptibilities than the thick and thin clouds that remain non-precipitating. Currently, we didn't find a possible explanation or related physical mechanism for this. We have modified the manuscript to clarify that the cloud memory hypothesis is less prominent for precipitating clouds transitioned from non-precipitating clouds.
  "Interestingly, as non-precipitating clouds transition to precipitating clouds (Fig. 10b and c, thin → rain, thick → rain), their LWP and $\alpha_c$ susceptibilities exhibit both less negative values and smaller daytime variations compared to thin/thick clouds that remain as thin/thick (Fig. 5b and c, thin → thin, Fig. 7b and c, thick → thick). The underlying reason for this observation is currently unclear and worth further investigations on the sensitivity of AIE for clouds experiencing transition in cloud states, especially between precipitating and non-precipitating clouds." (Lines 675-680).

- *Line 602, "instantaneous responses of warm boundary layer clouds…" I suggest rephrasing, as what you shown in the study is basically correlations (regression slopes), not actual responses (which refer to perturbation experiments).*
  Changed
- *Line 621, "instantaneous CF adjustment rate" is a bit awkward, I suggest rephrasing (see comment above).*
  Changed
- *Line 643, again, established mechanisms, not hypotheses.*
  Changed.
- *Font size too small in Figs. 3-7, 9-10. I suggest matching them to that of Fig. 11, which looks just fine.*
  Done.
- *In Fig. 3-7, 9-11, basically all time-series figures, I strongly recommend replacing the horizontal lines indicating daytime mean values with lines indicating "0", and one can simply label the daytime mean values on each panel. The current horizontal lines are so seductive to be seen as the zero lines.*
  Done.

**References**

[revised manuscript text omitted]

---

## Author Response (AR2)

**Responses to reviewer comments, round 2**
The original comments are in *blue italic font*, and our response are in black font.

**Reviewer #1:**
*The authors have carefully addressed all of my comments. Some minor comments below.*

*Lines in tracked changes document*
*L 554: verses > versus*
Changed.
*l. 555: than > rather than*
Changed.
*l. 564: this "memory" / adjusting to previous perturbations may make it more relevant to correlate the LWP/albedo at time t with the Nd at t – n hours… Do you know the work by Fons et al.?*
Thank you for the reference. Yes, this paper provides evidence on the temporal evolution of $N_d$ impact on both precipitation suppression feedback and entrainment-evaporation feedback from a causal inference perspective. This paper has been added to the refence.

*Ref: Fons, Emilie, et al. "Stratocumulus adjustments to aerosol perturbations disentangled with a causal approach." npj Climate and Atmospheric Science 6.1 (2023): 130.*

**Reviewer #2:**

*I greatly appreciate the efforts made by the authors to address my questions. Most of my concerns raised in the previous review have been sufficiently addressed.*
*After reading the revised manuscript, I have a few comments that I encourage the authors to consider:*
*1) Regarding CF susceptibility, your sensitivity test convinced me that the results you shown in Fig. 2c is qualitatively robust, but not quantitatively, as CF susceptibility is a function of cloud size. Moreover, the daytime evolution in CF susceptibility (Fig. 3c) suggests to me that not only the change of sign, but the determination of the sign of CF susceptibility is in general statistically insignificant. Could you indicate on the figures, especially for CF susceptibilities, when the value is statistically-significantly different from zero?*

Thank you for your suggestion. Figure 3 has been modified accordingly and shown below. As seen in Figs. 3b and c, the switch in sign for albedo susceptibility is statistically significant at a 95% confidence level, while the switch in sign for CF susceptibility is not statistically significant.

[Figure]

Figure 3. Daytime variation of cloud susceptibilities. (a) cloud LWP susceptibility ($dln(LWP)/dln(N_d)$), (b) cloud albedo susceptibility ($d\alpha_c/dln(N_d)$), (c) cloud fraction susceptibility ($dCF/dln(N_d)$), and (d) cloud shortwave susceptibility ($-dSW_{TOA}^{up}/dln(N_d)$). The shaded areas represent the lower and upper 25$^{th}$ percentile of the cloud susceptibilities for each time step and the solid lines with dots represent the mean values. In (b) and (c), filled

markers indicate data points that susceptibilities are significantly different from zero (p<0.05), while open markers indicate statistical insignificance.

*2) Since you investigated daytime cloud susceptibility evolution, I am surprised that shortwave absorption by cloud (a source for cloud dissipation) and its dependence on cloud LWP and Nd are not discussed at all when interpreting the results and formulating hypotheses. For example, thick clouds thin faster because of stronger SW absorption, compared to thin clouds. In other words, a key term in cloud LWP budgets during daytime is SW radiation, which is sensitive to LWP and Nd (e.g., Petters et al. 2012). I wonder how do SW absorption and its dependence on LWP and Nd affect your interpretation of the susceptibility evolution?*

We thank the reviewer for pointing out this important process. We have added related discussion to the manuscript: "In addition, clouds with higher $N_d$ and larger LWP exhibit stronger shortwave absorption, which enhance LWP depletion and therefore a more negative LWP susceptibility (e.g. Bores and Mitchell, 1994; Petters et al. 2012)." (Lines 387-389). "From late morning to early afternoon, with increasing solar radiation, deepening of boundary layer and clouds decoupled from surface, LWP susceptibility for thick clouds largely decreases and reaches a daily minimum, which contributes to the largest difference between the thin-to-thin and thick-to-thin categories shown in Fig.5b." (Lines 544-547). "From late morning to early afternoon, the overcast thick clouds break down and CF decrease with increasing $N_d$ likely due to the increased shortwave absorption, the enhanced entrainment, and evaporation." (Lines 653-655). "This is likely attributed to the stronger shortwave absorption, larger cloud top radiative cooling rate and stronger entrainment for thick clouds." (Lines 812-813).

*3) Regarding the "cloud memory of AIE or susceptibility" argument, my interpretation of your statement/hypothesis is still that clouds have memory of their past states, meaning their past rate of change in LWP (as in your words), which is governed by the environmental states they have been residing in for the past few hours. The separation between different cloud transition groups, such as thin to thin, thick to thin, etc., is probably set by boundary layer characteristics, e.g., BL depth and thermodynamics. Therefore, I think your classification of different cloud transition groups essentially represents different boundary layer conditions that dictate different cloud evolutions. I believe one can show this using large-scale meteorological conditions from reanalysis data.*
*To me, this is different from "cloud has a memory of its past susceptibility" as susceptibility in your study is simply a regression slope, and cloud cannot physically retain or "memorize" a statistical relationship. Therefore, I recommend rephrase how this hypothesis/argument is framed and discussed throughout the manuscript.*

Thank you for the comment. The text has been modified accordingly: "Therefore, we hypothesize that if clouds change state during the adjustment time, clouds may still retain the "memory" of their responses to $N_d$ perturbations from the previous state." (Lines 517). "In the afternoon, with increasing percentage of thick clouds develop from thin clouds and retain the memory of LWP responses to $N_d$ perturbations of the thin clouds." (Lines 649).

*Reference*
*Petters, J. L., J. Y. Harrington, and E. E. Clothiaux, 2012: Radiative–Dynamical Feedbacks in Low Liquid Water Path Stratiform Clouds. J. Atmos. Sci., 69, 1498–1512,*
*https://doi.org/10.1175/JAS-D-11-0169.1*